# Potent activity of polymyxin B is associated with long-lived super-stoichiometric accumulation mediated by weak-affinity binding to lipid A

**Kerry R. Buchholz** [1] ✉, **Mike Reichelt**[2], **Matthew C. Johnson**[3], **Sarah J. Robinson**[4], **Peter A. Smith**[1,6], **Steven T. Rutherford** [1] ✉ **& John G. Quinn** [5] ✉

Polymyxins are gram-negative antibiotics that target lipid A, the conserved membrane anchor of lipopolysaccharide in the outer membrane. Despite their clinical importance, the molecular mechanisms underpinning polymyxin activity remain unresolved. Here, we use surface plasmon resonance to kinetically interrogate interactions between polymyxins and lipid A and derive a phenomenological model. Our analyses suggest a lipid A-catalyzed, three-state mechanism for polymyxins: transient binding, membrane insertion, and super-stoichiometric cluster accumulation with a long residence time. Accumulation also occurs for brevicidine, another lipid A-targeting antibacterial molecule. Lipid A modifications that impart polymyxin resistance and a non-bactericidal polymyxin derivative exhibit binding that does not evolve into long-lived species. We propose that transient binding to lipid A permeabilizes the outer membrane and cluster accumulation enables the bactericidal activity of polymyxins. These findings could establish a blueprint for discovery of lipid A-targeting antibiotics and provide a generalizable approach to study interactions with the gram-negative outer membrane.

The outer membrane of gram-negative bacteria is an essential feature that serves as a permeability barrier to exclude toxigenic molecules, including antibiotics, and to provide physical rigidity to the cell[1–5]. The principal lipid component of the outer leaflet of the outer membrane is lipopolysaccharide (LPS), a complex glycolipid composed of a conserved lipid A anchor, a solvent-facing core oligosaccharide, and a variable, extracellular O-antigen[6,7]. These three components combine to produce a dense electronegative hydrogel barrier[8]. An interlinked LPS affinity network is established by hydrophobic packing of the acyl chains of lipid A and electrostatically stabilized by divalent metal ions[4,6]. Metal ion depletion and alterations to lipid A that negate electrostatic bridging are associated with outer membrane permeability due to loss of charge complementarity, which causes increased electronegative repulsion between adjacent LPS[4].

Lipid A is the target of the polymyxin class of antibiotics, which include polymyxin B and colistin, last-resort therapeutics for the treatment of multidrug resistant gram-negative bacterial infections[9–11]. These electropositive amphiphilic antibiotics can establish interactions with both the hydrophobic anchor and polar sugars of LPS and the associated cooperativity leads to enhanced binding avidity[9,10]. Polymyxin binding to lipid A leads to aggregation, disruption of outer membrane integrity, and, ultimately, bacterial death[12–14].

[1]Department of Infectious Diseases, Genentech, Inc., South San Francisco, CA, USA. [2]Department of Pathology, Genentech, Inc., South San Francisco, CA, USA. [3]Department of Structural Biology, Genentech, Inc., South San Francisco, CA, USA. [4]Department of Discovery Chemistry, Genentech, Inc., South San Francisco, CA, USA. [5]Department of Biochemical and Cellular Pharmacology, Genentech, Inc., South San Francisco, CA, USA. [6]Present address: Revagenix, Inc., San Mateo, CA, USA. ✉e-mail: buchholz.kerry@gene.com; rutherford.steven@gene.com; quinn.john@gene.com

The spread of polymyxin resistance, mediated principally through covalent modifications to lipid A, threatens the use of these clinically important antibiotics. Mutations in two-component regulatory systems, including *phoPQ*, *pmrAB*, and *basRS*, lead to increased production of inner membrane enzymes that covalently attach 4-amino-4-deoxy-L-arabinose (L-ara4N) or phosphoethanolamine (pEtN) to the phosphate groups of lipid A[15,16]. pEtN transferases can be encoded by *mcr* genes found on mobile plasmids[17]. Moreover, in *Vibrio cholerae*, lipid A can be modified with glycine and diglycine[18]. Each of these modifications alter the chemical nature of lipid A and are purported to disrupt polymyxin binding[15]. In the extreme, complete loss of LPS production can render strains of *Acinetobacter baumannii* resistant to polymyxins[19].

Although binding to lipid A in the outer membrane is essential for the effects of polymyxins, this alone does not explain their antibacterial activity. Variants of polymyxin B, specifically polymyxin B nonapeptide (PMBN), that do not possess antibacterial activity, nonetheless bind to lipid A and permeabilize the outer membrane to antibiotics that are normally excluded[20,21]. Likewise, modifications to lipid A that impart resistance do not prevent polymyxins from binding to and permeabilizing the outer membrane[22]. Evidence suggests that polymyxin B destabilization of membrane curvature upon lipid A binding is critical to its activity[23–25] and it has recently been proposed that polymyxin B targets lipid A both in the outer membrane, permeabilizing this barrier, as well as in the cytoplasmic membrane, causing cell lysis and death[26], however, the molecular mechanisms remain to be determined.

Investigations into the binding of polymyxins to bacterial cells have generally relied on whole-cell and plate-based equilibrium approaches. Different methods have produced equilibrium binding constants ($K_D$s) for polymyxin B ranging from 400 nM up to greater than 100 μM[27–31], which are, in some cases, multiple orders-of-magnitude higher than the reported cellular potency of these antibiotics. However, these approaches are likely complicated by the rapid kill-kinetics of polymyxins, non-specific binding of polymyxins to plate surfaces, assumptions of a simple binding mechanism, an inoculum effect on activity, the inability to reach equilibrium under the assay conditions, and the need for perturbative labels on the polymyxins[28,31–39]. Surface plasmon resonance (SPR) has been employed to measure polymyxin binding to LPS films, but has not been systematically applied to reveal the binding mechanism[40].

A mechanistic understanding of polymyxin B binding to lipid A could provide insight into how antibiotics interact with and penetrate through the outer membrane barrier and inform the design of antibacterial compounds able to overcome resistance. Here, we explore the binding mechanism of polymyxin B to bacterial cells, outer membrane vesicles (OMVs), and pure LPS using a variety of SPR-based assay formats augmented with microbiology techniques. Systematic kinetic interrogation of these complex interactions allowed derivation of a three-state mechanistic model. This model provides insight into the mechanism of action for this clinically important class of antibiotics, including the impact of lipid A modifications that cause resistance, and can potentially be applied to develop other lipid A-targeting antibiotics.

## Results
### Binding of polymyxin B to the outer membrane measured by SPR

Polymyxin B targets the conserved lipid A anchor of LPS and exhibits potent and selective gram-negative antibacterial activity[41]. This was confirmed by determining the minimal inhibitory concentration (MIC), or the concentration of compound required to completely inhibit bacterial growth, for polymyxin B. An MIC of 0.08 nM for polymyxin B was measured against wild-type *E. coli*, a model gram-negative species, whereas this antibiotic had no activity at the highest tested concentration against a gram-positive strain, *Staphylococcus aureus*, which lacks lipid A and an outer membrane (Table 1).

**Table 1 | Minimum Inhibitory Concentrations (MICs) of polymyxin B and rifampicin potentiation against wild-type and polymyxin-resistant strains**

| Species | Background | MIC (μM)[a] | | |
|---|---|---|---|---|
| | | Polymyxin B | PMBN[b] | Brevicidine |
| *E. coli* | WT | 0.00008 | >40 | 0.313 |
| | PmrA[G53E] | 10 | >40 | 0.625 |
| | *mcr*1[c] | 2.5–5 | >40 | 0.313 |
| | WT+rifampicin[d] | <0.00002 | 0.078 | 0.078 |
| | PmrA[G53E]+rifampicin[d] | 0.001 | 20 | 0.078 |
| | *mcr*-1+rifampicin[d] | 0.001 | 2.5 | 0.078 |
| *S. aureus* | WT | >40 | >40 | >40 |

[a]Minimal Inhibitory Concentrations (MICs): lowest concentration of antibiotic that completely inhibits bacterial growth.
[b]PMBN, polymyxin B nonapeptide.
[c]Strain carrying a plasmid encoding the *mcr*-1 gene.
[d]Rifampicin present at 1.56 μM, which has no effect on growth of *E. coli* or polymyxin-resistant *E. coli* strains tested.

To characterize the interactions between polymyxin B and LPS in the outer membrane, we employed SPR to monitor binding to whole *E. coli* cells or outer membrane vesicles (OMVs) linked to a planar surface (Fig. 1A). OMVs are naturally produced membrane spheres with diameters of 20–250 nm that capture the biological complexity of the gram-negative outer membrane[42,43]. Though the exact composition of OMVs can vary from species-to-species and under different culture conditions, they generally capture both the protein and lipid constituents of the outer membrane. OMVs present the exposed LPS layer in a biologically relevant orientation, and when OMVs were treated with polymyxin B they exhibited vesiculation and tubules, similar to the membranous perturbations observed on polymyxin-treated bacterial cells[23–25,42,44] (Fig. 1B and Supplementary Information (SI) Figs. S1–S3). Thus, OMVs represent a useful proxy for interrogating the interactions between polymyxins and LPS presented in a near-biological context by SPR.

The sensitivity of SPR decays exponentially with distance from the planar surface and is reduced to about 30% at an illumination wavelength of 270 nm. This is sufficient to detect binding to both whole *E. coli* cells, which have a diameter of approximately 0.5–1 μm along their shorter axis[45], and OMVs, which have diameters of 20–250 nm[42] (Fig. 1A). SPR optically probes the volume within this sensitivity depth to produce an averaged refractive index change, represented as response units (RUs), that is proportional to a change in concentration. The response is also weakly sensitive to mass redistribution within this volume because detection sensitivity decays exponentially from the surface[46]. Membrane vesicles from Expi293 cells[47] (Supplementary Information Fig. S4A) and OMVs from the model gram-negative bacteria *E. coli* (Supplementary Information Fig. S4B) were immobilized non-specifically on the surface of a lipophilic chip, and OMVs were immobilized via amine-coupled polymyxin B to the surface of a C1 chip (SI Fig. S4D) which allowed for better resolution of lower concentrations of polymyxin B (Supplementary Information Fig. S4C, S4F, S4H, and SI Methods). Electron microscopy showed discretely bound wild-type OMVs (wt-OMVs) that retained their spherical shape when attached to the planar chip surface (Fig. 1C).

To measure the interactions of polymyxin B with immobilized vesicles and cells, we employed single-cycle kinetic (SCK) injection and multicycle injection formats[48]. SCK is often preferred in cases where there is an accumulation of long-lived bound species as it is possible to obtain a full dose-response range in a single binding curve. SPR format, contact time, and dosing regimen were optimized to allow the complex kinetic processes of polymyxin B binding to be resolved (Figs. 1D and 2A).

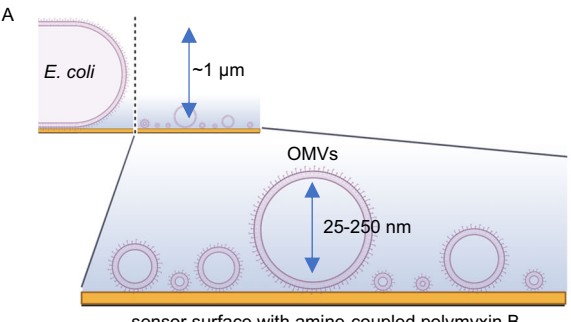

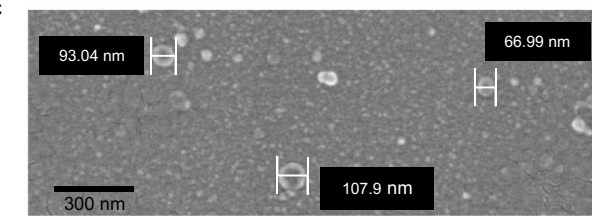

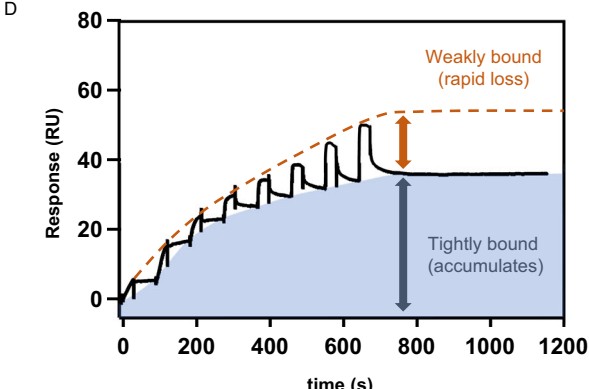

**Fig. 1 | SPR-detection of polymyxin B binding to the gram-negative outer membrane. A** Cartoon of SPR chip surface with bound whole bacterial cells or outer membrane vesicles (OMVs) (created with BioRender.com released under a Creative Commons Attribution-NonCommercial-NoDerivs 4.0 International license). **B** Transmission electron micrograph of wild-type OMVs (wt-OMVs) exposed to buffer (left) or polymyxin B (right). Representative images based on $n = 3$ biological replicates shown. Scale bar = 100 nm. **C** Scanning electron micrograph of wt-OMVs immobilized via amine-coupled polymyxin B to the surface of a C1 chip. Representative images based on $n = 3$ biological replicates shown. Scale bar = 300 nm. **D** SPR binding curves showing a stepwise increase in response (RU) upon exposure of wt-OMVs to eight serial-doubling concentrations of polymyxin B from 39 nM–5 μM in PBS plus 0.0005% tween-80. The tight binding (gray arrow) and reversible binding (red arrow) fractions are indicated and are observed in an approximately 2:1 ratio. SPR curve is representative of $n > 3$ independent replicates.

Binding response curves for serial injection of increasing concentrations of polymyxin B (Figs. 1D and 2A) or colistin (Supplementary Information Fig. S4E) over captured wt-OMVs revealed a binding profile dominated by the accumulation of tightly bound polymyxin B and a superimposed saw-tooth profile associated with transient binding. This saw-tooth profile was less apparent for low-concentration injections, likely because any polymyxin B-lipid A complexes that formed appear to rapidly transition to the tightly bound state that accumulated. Therefore, the observed curvature at low concentrations reflects the effective rate of accumulation of the tightly bound complex. At high concentrations, the binding capacity for tightly bound polymyxin B approaches saturation, allowing weak polymyxin B binding to OMVs to dominate with its characteristic saw-tooth profile. At saturation, the tightly bound component accounted for a higher proportion of total binding, approximately 2:1 over the saw-tooth component (Figs. 1D and 2A).

## Modifications to lipid A and polymyxin B alter the OMV binding profile

Covalent modifications of lipid A phosphate groups, including pEtN and L-ara4N, lead to polymyxin resistance, presumably by disrupting binding[49] (Supplementary Information Fig. S5). We confirmed that *E. coli* with a mutation in *pmrA* (PmrA[G53E]) or carrying a plasmid with an inducible *mcr*-1 gene (pBAD-*mcr*-1) were resistant to killing by polymyxin B (Table 1). As previously observed[20,22], polymyxin B still potentiated the activity of an antibiotic normally excluded by the outer membrane, rifampicin, against these strains, consistent with outer membrane disruption (Table 1). OMVs isolated from polymyxin-resistant *E. coli* PmrA[G53E] (resistant-OMVs) were composed of modified lipid A (Supplementary Information Tables S1 and S2 and Supplementary Information Fig. S6). Compared to wt-OMVs (Fig. 2A), serial injection of increasing concentrations of polymyxin B over captured resistant-OMVs exhibited only the weak, saw-tooth binding (Fig. 2B and Supplementary Information Fig. S4G).

The same polymyxin B binding profiles were observed with *E. coli* wild-type (Fig. 2C) as well as polymyxin-resistant *E. coli* pBAD-*mcr*-1 (Fig. 2D) and *E. coli* PmrA[G53E] (Supplementary Information Fig. S4I) whole cells, indicating OMVs are a reliable model for studying polymyxin B interactions with the outer membrane by SPR. As a tool, OMVs reduced experiment-to-experiment variability, likely caused by stochastic population differences among growing bacterial cells, eliminated compounding biological considerations, and enabled more consistent and sufficient levels of capture on the SPR chip surface for quantitative analyses.

A polymyxin B variant lacking a terminal amino acid and acyl tail, polymyxin B nonapeptide (PMBN) (Supplementary Information Fig. S7), also lacks the antibacterial activity of polymyxin B but still permeabilizes the outer membrane to rifampicin[20,21] (Table 1). Serial injection of increasing concentrations of PMBN over wt-OMVs (Fig. 2E), resistant-OMVs (Fig. 2F), and bacterial cells (Supplementary Information Fig. S8) exhibited only weak, saw-tooth binding. To directly compare the saw-tooth binding fractions, RUs were converted to concentrations, normalizing for molecular weight, and thus revealing that the saw-tooth fraction binding capacity of OMVs is similar across conditions except at the lowest concentrations of polymyxin B where stable binding dominates (Supplementary Information Fig. S9).

## Divalent cations disrupt polymyxin B binding to OMVs

Divalent metal cations stabilize the LPS network and compete with polymyxins for the negatively charged phosphates of lipid A[4,50]. To determine if excess divalent cations compete with or limit binding of polymyxin B to OMVs, we performed SCK binding in the presence of excess magnesium ions ($Mg^{2+}$). With 32 mM added $Mg^{2+}$, the interaction of polymyxin B with OMVs was reduced to a weak transient saw-tooth profile without stably bound polymyxin B, and the interaction of

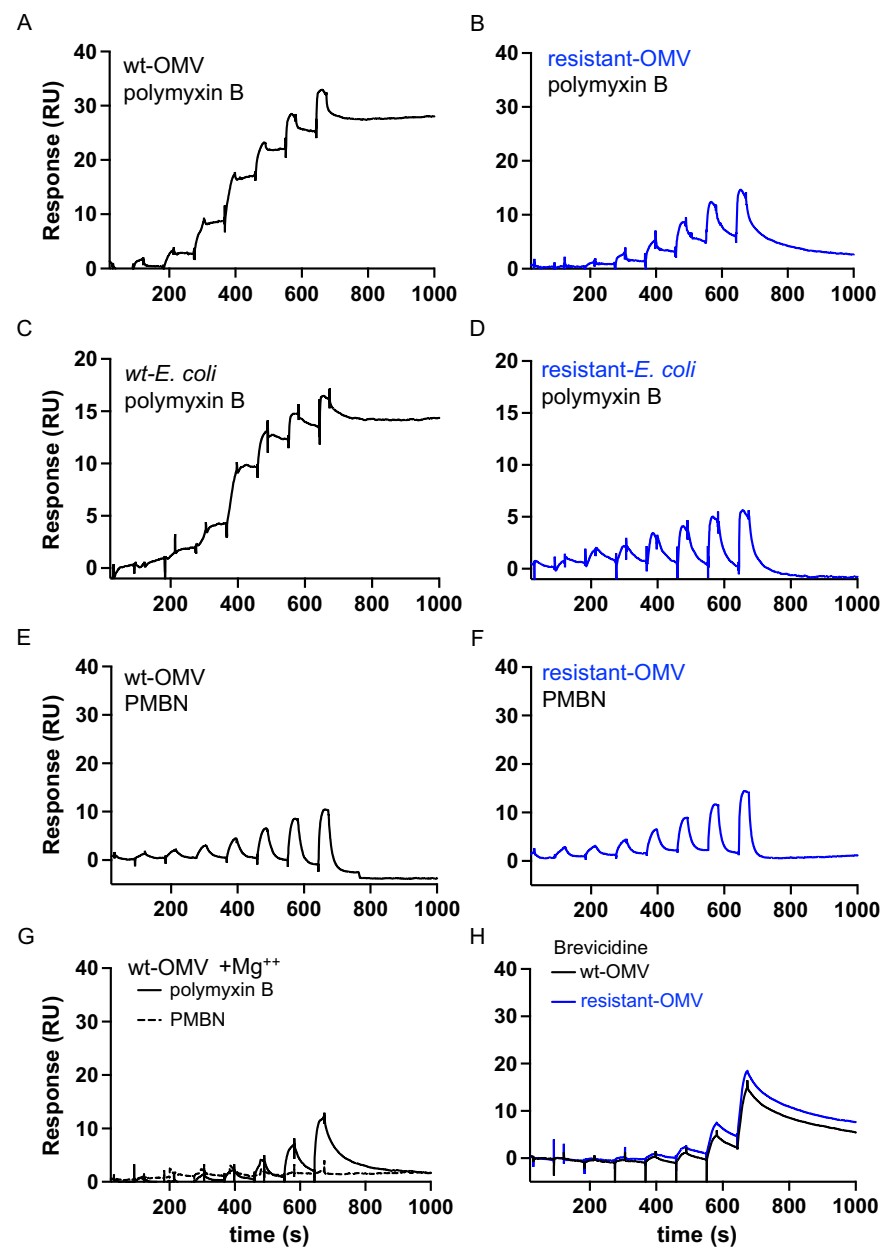

**Fig. 2 | SPR curves for binding of antibiotics to whole bacterial cells or OMVs.** Each serial injection profile was recorded as outlined in Fig. 1 with eight serial doubling concentrations with a maximum concentration of 625 nM. Representative SPR sensorgram of polymyxin B binding to (**A**) wt-OMVs (253 RU loaded) or (**B**) resistant-OMVs isolated from *E. coli* PmrA[G53E] cells (250 RU loaded). Representative SPR sensorgrams of whole cell binding by polymyxin B with (**C**) wild-type *E. coli* cells (169 RU loaded) or (**D**) polymyxin B-resistant *E. coli* cells expressing *mcr*-1 (102 RU loaded). Representative SPR sensorgrams of PMBN binding with (**E**) wt-OMVs

(512 RU loaded) and (**F**) resistant-OMVs isolated from *E. coli* PmrA[G53E] cells (681 RU loaded). **G** Representative SPR sensorgrams of polymyxin B (solid line, 245 RU loaded) or PMBN (dashed line, 206 RU loaded) binding to wt-OMVs in the presence of 32 mM Mg[2+]. **H** Representative SPR sensorgrams of brevicidine binding to wt-OMVs (black line, 462 RU loaded) or resistant-OMVs isolated from *E. coli* PmrA[G53E] cells (blue line, 422 RU loaded). All traces are representatives of *n* ≥ 3 independent SPR runs.

PMBN with OMVs was eliminated (Fig. 2G). Moreover, EDTA, a metal chelator that disrupts outer membrane permeability by removing divalent cations[4], had no effect on polymyxin binding (Supplementary Information Fig. S10). These findings are consistent with a role for electrostatic interactions in both the association of polymyxin B with the outer membrane and, potentially, subsequent steps.

### Brevicidine binding to OMVs measured by SPR
Brevicidine was recently identified as a cationic, non-ribosomal, natural product peptide with selective activity against gram-negative bacteria due to lipid A targeting[51]. MICs of 0.313 μM

against *E. coli* and >40 μM against *S. aureus*, confirmed this activity profile (Table 1). Brevicidine exhibited a complex interaction with wt-OMVs composed of accumulation of a tightly bound species and a superimposed saw-tooth profile (Fig. 2H), similar to the binding profile observed for polymyxin B (Fig. 2A), though slow dissociation was observed for brevicidine but not polymyxin B. Unlike polymyxin B, brevicidine exhibits antibacterial activity against strains with polymyxin-resistant lipid A modifications[51] (Table 1). Strikingly, and consistent with its antibacterial activity, brevicidine binding to polymyxin-resistant-OMVs was indistinguishable from its binding to wt-OMVs (Fig. 2H).

## Measuring apparent rate constants for polymyxin B

The off-rate of the tightly bound fraction was estimated using a chaser analysis[52], where OMVs were pre-saturated with polymyxin B and then re-saturated after an extended time interval (Supplementary Information Fig. S11A). The amount of accumulation upon re-saturation relative to the initial saturation step revealed the loss in occupancy and allowed calculation of a residence time ($\tau = 1/k$) and half-life for the interaction ($t_{1/2} = \ln2/k$)[52]. A $t_{1/2}$ of >6 h was determined for polymyxin B binding to OMVs at both 25 and 37 °C (SI Table S3). The $t_{1/2}$ of brevicidine, also measured by chaser analysis, was approximately 1 h on both wt-OMVs and polymyxin-resistant-OMVs (SI Table S3).

An apparent-$K_D$ was measured for the reversible saw-tooth binding profile where polymyxin B was titrated over an OMV surface pre-saturated with the stable binding component (Supplementary Information Fig. S11B). The resulting binding curve displayed steady-state regions at the maxima of each saw-tooth profile that could be taken as steady-state responses and enabled a simple affinity isotherm fit. No pre-saturation was necessary when using resistant-OMVs or PMBN as these lack the stable component (Fig. 2B, D–F). The change in RUs from the base to the plateau of each step was used to construct a standard affinity plot that enabled determination of an apparent-$K_D$ of 517 nM for the reversible, saw-tooth binding of polymyxin B (Supplementary Information Table S4). The measured apparent-$K_D$s and apparent curvatures were not significantly different among the treatment groups, suggesting that with reversible, saw-tooth binding events for polymyxin B and PMBN with wt-OMVs and resistant-OMVs represent the same lipid A-binding mechanism.

## Mechanistic studies of polymyxins binding to cells, OMVs, and LPS

We performed additional SPR experiments to measure binding of polymyxin B and PMBN to *E. coli* cells, OMVs, and purified LPS under physiological concentrations of salts and divalent cations and at 37 °C. We summarize the mechanistic studies here (Fig. 3) and present a complete, detailed description (Supplementary Information Section 1).

Thus far, we have employed SCK injections as this produces information-rich titration curves over a range of compound concentrations without requiring surface regeneration and is often preferable for analysis of tightly bound compounds. However, multicycle kinetics (i.e., one concentration per sensorgram) might also be adopted for any compound and is preferable for reversible binding compounds as a full dissociation profile is obtained at each concentration tested, which benefits mechanistic modeling. The multicycle SPR curves of PMBN binding to whole cells and OMVs show kinetic curvature that resembles binding kinetics but is instead caused by the development, and decay, of a mass transport-limited boundary, reporting the binding reaction at quasi-steady-state (Fig. 3A, B; SI Section 1A and Supplementary Information Section 1B). The high-quality fit to the boundary layer model (Supplementary Information Section 1A - Eq. (S3)) confirms that under these conditions, PMBN (and polymyxin B as described below) binds rapidly despite the presence of divalent cations. A finite element based-numerical model (Supplementary Information Figs. S12 and S13) shows that such mass transport dominance also applies to single cells, which is relevant to in vivo milieus.

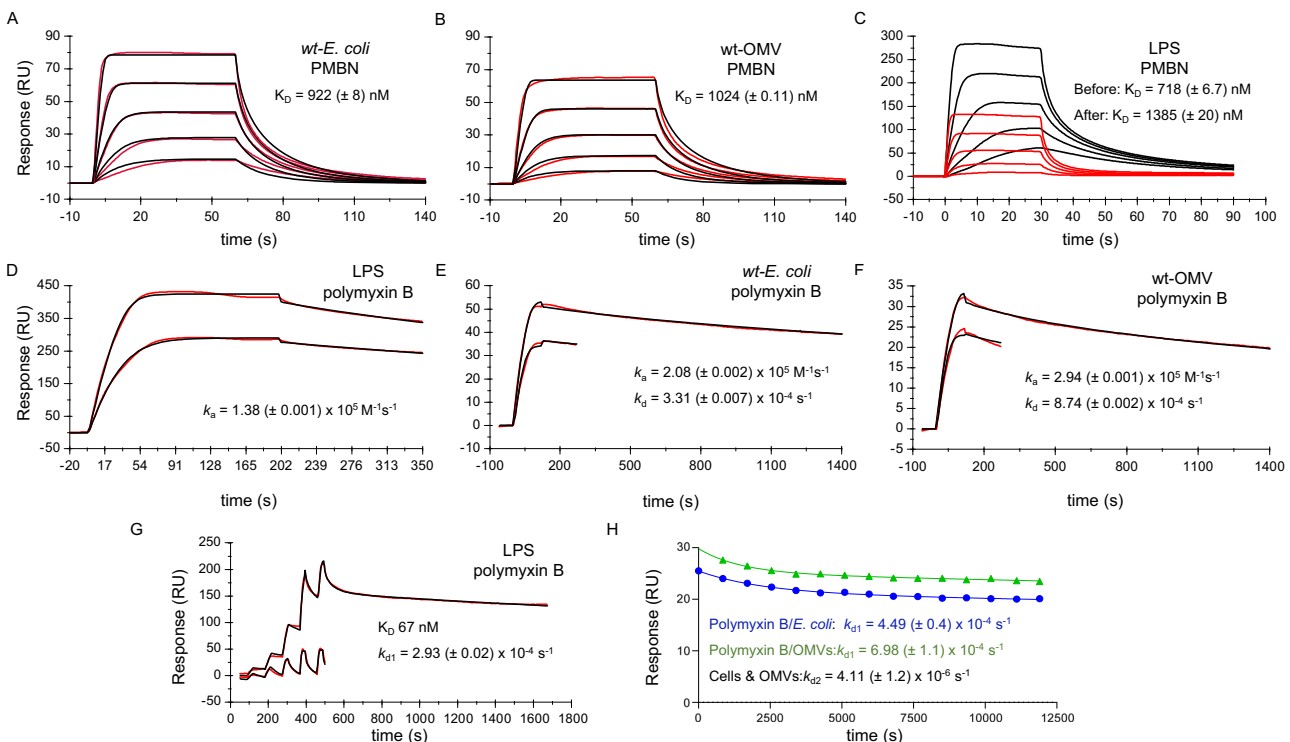

**Fig. 3 | Binding of polymyxin B and PMBN to *E. coli* cells, OMVs, and LPS and fitting to approximate kinetic models.** Affinity analysis of PMBN binding to (**A**) *E. coli* cells and (**B**) OMVs using a 1:1 kinetic boundary model fit (Eq. (S3)). **C** PMBN binding to LPS (black curves) and replicated (red curves) after pre-saturation of the surface with polymyxin B, as shown in (**D**). Both curve sets were fit to Eq. (S3) to estimate affinity. **D** Saturation of the LPS surface with polymyxin B repeated at two different LPS densities and fit to a simple 1:1 kinetic binding model. Binding of polymyxin B to (**E**) *E. coli* cells, and (**F**) OMVs fit to a 1:1 two-compartment binding model (Eq. (S2)). **G** Upper curve: binding of polymyxin B to LPS and fit to a two-state kinetic binding model (Eq. (S6)). Lower curve: replicate of upper curve where the LPS surface was pre-saturated with polymyxin B as shown in (**D**). **H** Polymyxin B dissociation curves for cells and OMVs pre-saturated with polymyxin B. Polymyxin B occupancy was obtained by chaser SPR analysis, where repeated PMBN injections report changes in polymyxin B occupancy allowing dissociation to be estimated at multiple timepoint readings that were then fit to a two-state dissociation model (Eq. (S7)) and Supplementary Information Sections A and D).

We next sought to investigate the binding mechanism that produces both reversible states and more stable polymyxin B bound states by comparing differences in affinity, binding capacity, and stoichiometry after full saturation with polymyxin B (Fig. 3C, D, and in more detail in SI Fig. 15). The affinity constant and binding capacity for PMBN binding were estimated from fitting Eq. (S3) (Supplementary Information Section 1A) and indicated a two-fold drop in both affinity and binding capacity when LPS was pre-saturated with polymyxin B. This implies that only a fraction of polymyxin B binding events can transition to a more stable bound state(s) ($t_{1/2}$ of >6 h (Supplementary Information Table S3)). Polymyxin B binding to LPS exhibited exponential binding kinetics towards a defined saturation limit (Fig. 3D) that is consistent with high occupancy of available LPS.

Direct binding of polymyxin B to *E. coli* cells, OMVs, and LPS (Fig. 3E–G) showed tight polymyxin B binding for all three targets. Indeed, the apparent association rate constant, which is driven by transport kinetics and binding affinity, when averaged for cells, OMVs, and LPS was in good agreement ($k_a$ 2.1 ($\pm$0.78) $\times 10^5$ M$^{-1}$s$^{-1}$). At moderate dissociation times (<1400 s), the dissociation is dominated by a moderate rate that was relatively consistent (<3-fold variation) when averaged over all three targets. Tightly bound polymyxin B was observed over LPS (Fig. 3G, high response curve) and again the saturation capacity for the tightly bound component was well below full LPS saturation allowing tightly bound polymyxin B and transiently bound polymyxin B to co-exist, which is observable as the additional reversible saw-tooth-shaped binding profile. Interestingly, repeated serial injections of polymyxin B after pre-saturating the LPS surface with polymyxin B isolated the reversible, transient binding component alone (Fig. 3G, low response curve, and Supplementary Information Fig. S14), implying that saturation of the tightly bound polymyxin B component is limited by another process. Indeed, a PMBN-like binding profile was observed which also resembled resistant-OMVs and resistant *E. coli* cells exposed to polymyxin B (in the absence of pre-saturation), implying that affinity of polymyxin B for LPS can be estimated (Supplementary Information Fig. S14) by fitting the boundary layer model (Supplementary Information Section 1A - Eq. (S3)).

We determined dissociation kinetics using a single-point chaser method (Supplementary Information Table S3), and next measured dissociation of polymyxin B from cells and OMVs using a more rigorous multipoint chaser method as described above[52], where repeated PMBN injections report changes in polymyxin B occupancy allowing dissociation to be estimated without interference from baseline drift (Fig. 3H). This analysis revealed a biphasic dissociation where a practically irreversible component was observed relative to the doubling time of bacterial growth. Apparent binding kinetics to cells and OMVs were nearly identical and binding/unbinding to purified LPS deviated by just 2-fold. Overall, the agreement between all three LPS containing surfaces suggests that OMPs and other components of the cell membrane are not required for prolonged retention of polymyxin B. Dissociation of polymyxin B species is heterogeneous due to the presence of nucleates and clusters, which is apparent over short time-courses. However, clusters predominant in the outer membrane over long time-courses. To isolate the apparent retention time for each dissociation phase, it is important to measure the dissociation at multiple time points over a prolonged period of several hours (Fig. 3H). Importantly, the apparent association and dissociation rate constants reported in Fig. 3E, F lump all bound states into a single state while those in Fig. 3G, H assume two states. These models are mechanistically over simplistic and ignore the dominance of mass transport, yet they do allow qualitative comparison of observed kinetic curvature that imply a high probability of shared mechanism.

## Three-state cluster model for polymyxin

Approximate kinetic/affinity modeling methods (Supplementary Information Section 1) revealed that binding of polymyxin B might be more fully elucidated by developing a mechanistic interaction model that can be fit directly to complex SPR data sets. The reaction mechanism for polymyxin B loading derives from the complex thermodynamics of the system and is encoded in the observed kinetic curvature. Formulating a phenomenological mechanistic model that describes the complex kinetics informs mode-of-action to benefit antibiotic discovery. The number of kinetic transitions that were required to robustly fit the experimental data was iteratively minimized resulting in three polymyxin B-bound states (Fig. 4).

This three-state mechanistic model follows mass conservation allowing the fraction of polymyxin B contained in each state to be estimated. It defines the kinetic evolution of the system without requiring a structural understanding of each microscopic state. A full physical understanding of each state is beyond the scope of the current work, but we nevertheless provide a possible interpretation of the

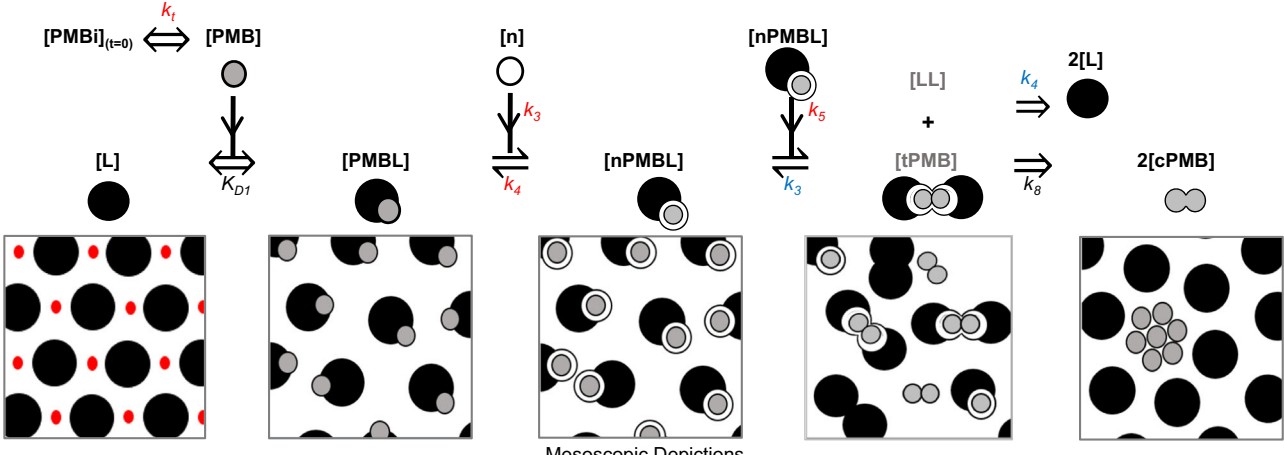

**Fig. 4 | Reaction mechanism for LPS-targeting antibiotics.** A proposed structural model that describes LPS-catalyzed super-stoichiometric accumulation of polymyxin B clusters, **cPMB** mediated by transition state intermediates is shown. **PMBi** is injected polymyxin B; **PMB** is polymyxin B (gray circle); **L** is LPS (black circle), divalent metal ions are red circles; **PMBL** is a transient polymyxin B-LPS complex; **n** indicates available membrane insertion sites (open circles); **nPMBL** is the membrane-inserted polymyxin B-LPS species; **tPMB** is a transient **nPMBL** dimer and **LL** is a transient LPS dimer; and **cPMB** is a polymyxin B monomer within a cluster. $k_t$ is a mass transport rate; $K_{D1}$ is an affinity constant for mass transport-limited binding. Rate constants in red fonts were used when fitting this model to polymyxin B binding to wt-OMVs in Fig. 5A and rate constants in blue fonts were repeating rate constants that appear in other reaction steps.

model (Fig. 4, detailed in Supplementary Information Section 2B) which will likely require refinement as a physical understanding becomes available. The model describes lipid A-catalyzed accumulation of polymyxin B clusters in the outer membrane, **cPMB**, potentially through phase separation mediated by transition state intermediates. The reaction pathway is mechanistically precise, however, the illustrations (Fig. 4) depict a clustering mechanism that may be driven by phase separation that has yet to be confirmed structurally. It is useful to adopt this concept of clustering via phase separation as a provisional basis for interpretation of the observed interactions.

While SPR has been used extensively for evaluating affinity binding, it can also be applied to phase transitions[53] because a change in phase state will be accompanied by a change in stability relative to the previous state which is observable in the kinetic curvature measured by SPR. The true mechanism is therefore already encoded in the curves and the objective of the iterative modeling process is to discover this mechanism. An LPS affinity network consisting of an LPS (**L**) monolayer stabilized by divalent cations (red dots) is exposed to polymyxin B (**PMB**). Advection and diffusion transport the injected polymyxin B (**PMBi**) to the SPR sensing surface at mass transport rate $k_t$, where it will attain concentration [**PMB**] (gray circles) and react with **L** to produce a transient complex, **PMBL**, which is the first bound state. Formation of **PMBL** competitively displaces divalent metal ions, which lowers the stability of the LPS-metal ion affinity network. The affinity constant ($K_{D1}$) replaces the associated transient kinetic rate constants because binding is fully mass transport-limited (see Supplementary Information Section 1A). The weakened LPS-affinity network increases the availability of membrane insertion sites, **n**, associated with each LPS molecule. **PMBL** interacts lipophilically with **n** to produce a membrane-inserted species **nPMBL** which is the second bound state. The model assumes that all polymyxin B-bound states are described on a monomer basis other than the transition state intermediates that require a dimeric state to trigger a proposed phase separation to the third state, **cPMB**. The transition state begins with self-association of **nPMBL** complexes through interactions between each respective polymyxin B contained in the dimeric **nPMBL**. These interactions displace pre-existing interactions between each **PMB** and its paired **L** thereby forming transient membrane-inserted polymyxin B dimers (**tPMB**) and LPS dimers (**LL**). The transition state intermediate **tPMB** is fundamentally a form of nucleate and, therefore, we might expect it would share the same dissociation constant ($k_4$). This was the case when fitting the model as dissociation of the **LL** intermediate state gated release of **cPMB** and matched the dissociation of **nPMB** from the acyl LPS matrix to form **PMBL**. The rate constant ($k_8$) for dissociation of **cPMB** from **tPMB** had no effect on the data and was therefore non-limiting and held constant at an arbitrary high non-limiting value (>1). **tPMB** does not accumulate significantly because it dissociates irreversibly into **cPMB**. When fitting to polymyxin B binding data for wt-OMVs (Fig. 5A), we held $K_{D1}$ and $k_8$ as constants while fitting rate constants $k_t$, $k_3$, $k_4$ and $k_5$ (red font) and found that for the resistant-OMVs the value of $k_5$ approached zero, effectively eliminating $k_5$ and $k_8$.

### Kinetic characterization of polymyxin B binding to OMVs

To estimate binding constants, the three-state model was fit to binding curve titrations (Fig. 5A, B, and Table 2). Briefly, the SPR response curves of polymyxin B binding wt-OMVs (Fig. 5A) or resistant OMVs (Fig. 5B) were globally fit to *Eqn. (1)* by non-linear regression coupled to numerical integration of the associated ordinary differential equations (see Supplementary Information Section 2A), which define the time-dependent concentrations of each species.

$$Response_{(t)} = ([\mathbf{PMBL}]_t + [\mathbf{nPMBL}]_t + 2*[\mathbf{tPMB}]_t + m.[\mathbf{cPMB}]_t).MW.G.$$

(1)

The bounded 2D fitspace 98% confidence contours[54] (Fig. 5C, D) indicate that all fitted parameters are well bounded and with reasonable confidence limits. Strongly elliptical contour boundaries indicate significant parameter correlation and as expected, the rate of formation of nucleates ($k_3$) over wt-OMVs and their rate of dimerization ($k_5$) are correlated (Fig. 5C), while $K_{D1}$ (Fig. 5D) is also correlated with formation of nucleates for resistant OMVs, though confidence limits remain acceptable in all cases. In the case of wt-OMVs, fitting $K_{D1}$ in addition to four other rate constants did result in excessive correlation and was avoided by pre-estimating affinity ($K_{D1}$) using a fit-for-purpose SPR format (see Supplementary Information Fig. S14) and then holding it constant at this value. Taken together the 2D fitspace analysis, parameter fit error (SE and confidence limits), and goodness of fit ($\chi^2$) provide high confidence in the three-state model as a reliable functional model for mechanistic analysis of antibiotics that follow a polymyxin-like, lipid A-targeting mode-of-action (Fig. 5A, B, and Table 2).

The analysis shows that polymyxin B binds wt-OMVs transiently, forming the first bound state, which would be expected from strong complementary electrostatics[55,56] and required adoption of a fully transport model (Supplementary Information Section 1A - Eq. (S3)) for this bound state. We pre-determine the apparent affinity ($K_{D1}$ = 126 nM (Table 2)) for formation of this bound state on wt-OMVs. Formation of **nPMBL** was moderate ($k_3$ = 2233 M$^{-1}$s$^{-1}$) as was the stability (residence time = $1/k_4$ = 205 s) of these nucleates. More importantly, both these rate constants are also associated with the transition state intermediates. The rate of dimerization of **nPMBL** to form the transition state intermediates was ~10-fold slower ($k_5$ = 20.75 M$^{-1}$s$^{-1}$) than formation of **nPMBL** while the reverse rate constant was matched. Similarly, the dissociation rate constant of **nPMBL** and recovery rate of free LPS were matched. These repeating rate constants are consistent with a nucleation process that generates transition state intermediates where both are limited by membrane stretching. Thus, bounded membrane stretching prevents clusters from accumulating at the diffusion limit, as is often observed[57]. The effective rate constants observed for polymyxin B binding to resistant-OMVs (Table 2) were considerably reduced and formation of the transition state was negligible ($k_5 \approx 0$) which prevented accumulation of long-lived clusters. This is made visually apparent in the species component plots where **cPMB** is increasingly dominant over time for wt-OMVs (Fig. 5A) and remains near zero for resistant-OMVs (Fig. 5B).

## Discussion

The importance of lipid A binding for the antibacterial activity of the polymyxins is well-established, but there is no clear consensus on the binding mechanism or how binding translates to outer membrane permeabilization and bactericidal activities. SPR enabled the resolution of polymyxin B binding kinetics from a second out to several hours. Mechanistic modeling of data from multiple SPR assay formats, using whole bacterial cells, OMVs, and pure LPS, resulted in a three-state model supporting the hypothesis that super-stoichiometric polymyxin B accumulation is necessary for the activity of lipid A-targeting antibiotics.

Briefly, the model was derived from a set of assumptions (see Supplementary Information Section 1I) generated from multiple mode-of-action experiments (see Supplementary Information Section 1B–F) and provides a basis to rationalize published observations on polymyxin B activity. For example, we initially observed that **cPMB** cluster accumulation was on the order of total LPS content, but a supply of membrane insertion sites was required to couple the formation of each species apparent from the SPR curvature. This excess of clusters is likely the trigger for formation of deformities in the outer membrane observed on OMVs and bacterial cells treated with polymyxin B[23–25] (Fig. 1B and Supplementary Information Figs. S1, S2). Additionally, the persistence of long-lived interaction in

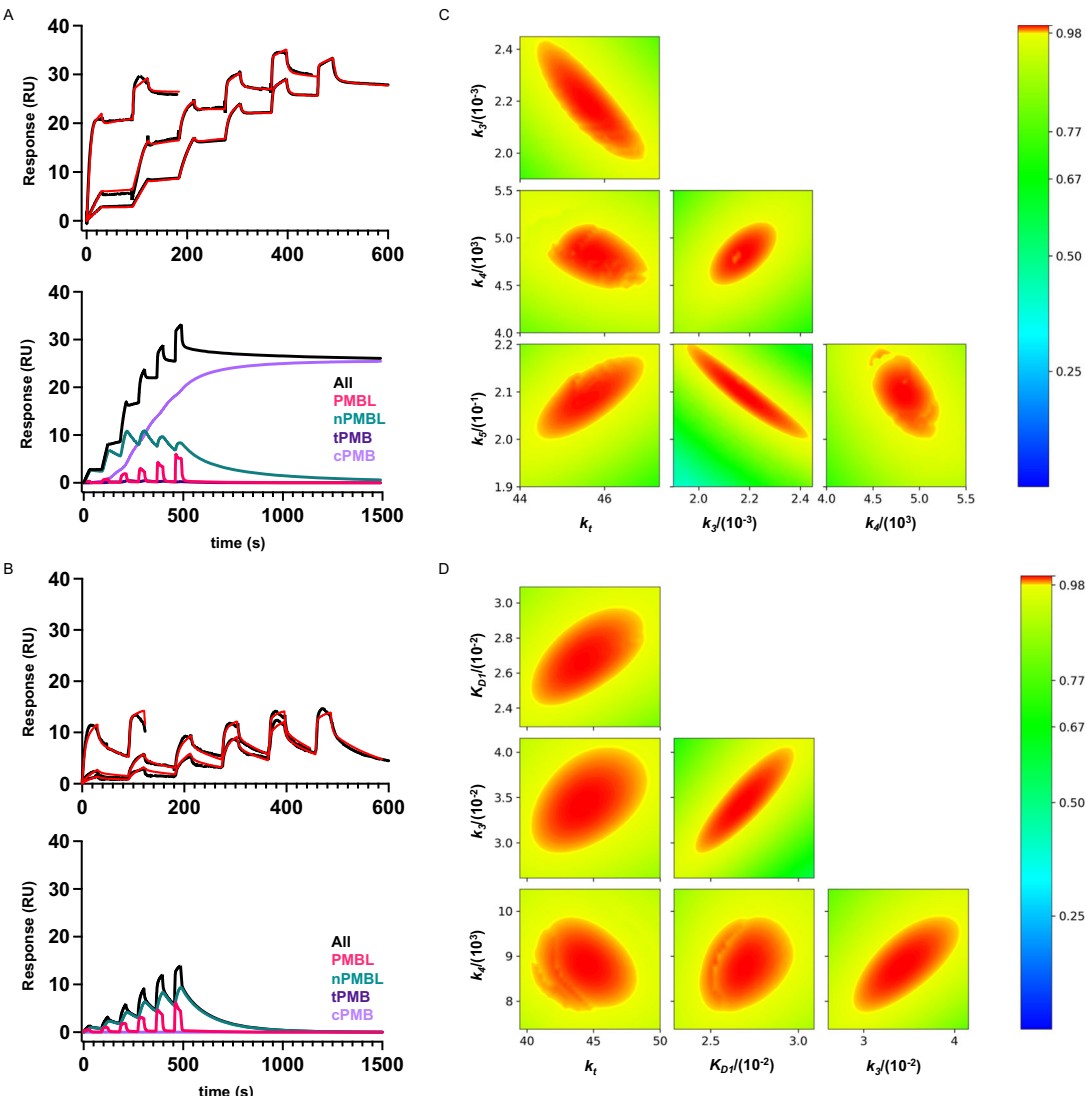

**Fig. 5 | Three-state model fitted to experimental SPR binding curves.** Titration of polymyxin B to a maximum concentration of 625 nM with (**A**) wt-OMVs and (**B**) resistant-OMVs. The fitted SPR curves are shown (left, upper panel - SPR data (black), modeled (red)) together with decomposition of one of these binding curves into component species (left, lower panels - PMBL (pink), nPMBL (turquoise), tPMB (dark purple), cPMB (light purple), composite (black)). The fitted model is near superimposable upon the experimental SPR binding curves. **C**, **D** 2D fitspace analysis associated with each fitted data set are shown (right panels). Binding constants were constrained to global values per curve set and the resulting parameter values, standard error associated with the fit, confidence intervals, and χ2 values are summarized in Table 2.

the outer membrane can explain the polymyxin inoculum effect, wherein MICs increase as the number of bacteria increase[58]. Polymyxin B in clusters are strongly retained in the outer membrane due to insolubility and do not freely diffuse into the aqueous phase. Therefore, upon cell division, polymyxin B clusters per cell will be reduced, presumably until the amount is insufficient for steps necessary for bacterial cell killing.

Overall, we propose that the interaction of polymyxin B with the outer membrane entails coupling of transient LPS binding to accumulation via a clustering mechanism resulting in formation of tightly retained polymyxin B clusters (Figs. 4 and 5). In the first step, polymyxin B binds to LPS, approximated as a simple 1:1 complex **PMBL**, with a relatively moderate apparent $K_D$ of approximately 126 nM when tested at nM concentrations. Formation of **PMBL** destabilizes the LPS network and disrupts the outer membrane barrier through displacement of divalent metal ions. In the next step, polymyxin B in **PMBL** can either unbind and dissociate from the outer membrane or undergo membrane insertion to form nucleates, **nPMBL**. The SPR data show

that nucleates remain stoichiometrically associated with lipid A and that **PMBL** and **nPMBL** co-exist. In the third step, accumulation of **nPMBL** further destabilizes the outer membrane by disrupting LPS packing, and this likely favors spontaneous self-association of **nPMBL** into larger, stable, lipid A-free polymyxin B clusters, **cPMB**. The lifetime of these clusters, $t_{1/2} > 6$ h, exceeds the doubling time of bacteria making them essentially irreversible on the timescale of growth. The model is consistent with "self-promoted uptake" wherein polymyxin B induces the lipid A layer to act as a catalyst that promotes accumulation of super-stoichiometric concentrations of polymyxin B into clusters. Importantly, weak polymyxin B binding to lipid A is necessary to allow subsequent dissociation to effectively trap polymyxin B into long-lived clusters.

The proposed three-state clustering model provides a functional interpretation for the mechanism of polymyxin B accumulation without a definitive understanding of its structural underpinnings. Therefore, future efforts aimed at defining the precise structural basis of these interactions, including the possible action of phase separation as

**Table 2 | Binding parameters for polymyxin B returned from fitting the three-state model (Fig. 5A, B) with global constraint of all interaction constants**

| | value[a] | ±SE of fit | 98% CI (profile likelihood)[b] |
|---|---|---|---|
| wt-OMV[c] | | | |
| $K_D = k_2/k_1$ (nM) | 126* | | |
| $k_3$ (M$^{-1}$ s$^{-1}$) | 2233 | 98 | 1950–2450 |
| $k_4$ (s$^{-1}$) | 0.00488 | 0.00007 | 0.00444–0.00524 |
| $k_5$ (M$^{-1}$ s$^{-1}$) | 29.76 | 0.02 | 20–22 |
| $k_t$ (ms$^{-1}$) | 45.3 | 0.1 | 44.3–47.1 |
| $\chi^2$ (RU$^2$) | 0.42000 | | |
| resistant-OMV[c] | | | |
| $K_D = k_2/k_1$ (nM) | 268 | 3 | 243–297 |
| $k_3$ (M$^{-1}$ s$^{-1}$) | 335 | 1 | 290–404 |
| $k_4$ (s$^{-1}$) | 0.0085 | 0.0002 | 0.0078–0.0099 |
| $k_t$ (ms$^{-1}$) | 44.6 | 0.4 | 40.4–48.8 |
| $\chi^2$ (RU$^2$) | 0.47 | | |

[a]Values generated calculated from the three-state model with $K_D$ in wt-OMVs (*) fixed to experimentally determined value for wt-OMV. For binding curves with modified LPS (resistant-OMV) the model defaulted to the first two-states.
[b]Confidence limits were calculated using the brute force method where confidence contours were generated by fitting all parameters to the actual data set while holding a given parameter constant and repeating with values to either side of the optimal fitted value of that parameter.
[c]OMVs isolated from wild-type or $pmrA^{G53E}$ polymyxin-resistant *E. coli* strains.

well as other potential interactions, will provide critical insight into the molecular mechanism of actions for these critical antibiotics.

Our model also provides insight into the differences in binding that occur when polymyxin B or lipid A are modified. PMBN lacks a hydrophobic tail and does not kill bacterial cells[9]. This polymyxin B variant exclusively produces rapidly reversible binding (Figs. 2E, 3A, and B). Strikingly, PMBN binding to OMVs is identical to polymyxin B binding when polymyxin B clusters are pre-saturated (Supplementary Information Fig. S9 and S11B) and this recapitulated with pure LPS (Fig. 3G), indicating that while PMBN can form **PMBL**, the lack of an acyl tail prevents further transitions. The absence of a lipophilic anchor reduces the amphiphilic properties of polymyxin B, thereby limiting lipophilic membrane interactions and promoting greater solubility. However, because PMBN can still permeabilize the outer membrane (Table 1), formation of **PMBL** must disrupt the LPS affinity network, likely through competition for metal ion binding sites.

Modifications to the phosphate groups of lipid A impart polymyxin B resistance[49], and we observed that these modifications eliminate cluster formation (Fig. 2B, D). The absence of **cPMB** clusters observed with resistant-OMVs may be related to an electrostatically enhanced stability of the affinity network that can more effectively resist the outer membrane stretching that is likely needed for accumulation of clusters. Presumably, formation of small nucleates, **nPMBL**, which were observed for resistant-OMVs (Fig. 2B), does not require a high degree of energetically costly membrane stretching. We speculate that cluster formation could promote cell killing by enabling a transmembrane flux of polymyxin B at the stretched phase boundary of cluster sites, though this remains to be directly demonstrated.

We explored the generalizability of our model by monitoring binding of brevicidine, a distinct lipid A-binding antibacterial natural product[51]. Brevicidine exhibits a binding pattern similar to that observed for polymyxin B with SPR traces showing accumulation of a tightly bound species and a superimposed saw-tooth profile associated with transient binding (Fig. 2H), suggesting polymyxin B and brevicidine might exploit a similar binding mechanism when interacting with lipid A in the outer membrane. The complex formed by

brevicidine ($t_{1/2} > 1$ h) was less stable than those formed by polymyxin B ($t_{1/2} > 6$ h) (SI Table S3) and this could account for the less potent antibacterial activity of brevicidine (Table 1). Also distinct from polymyxin B, brevicidine displayed an identical response to wt-OMVs and resistant-OMVs (Fig. 2H), consistent with the antibacterial activity of brevicidine against polymyxin-resistant mutants (Table 1). Thus, though the molecular interactions with lipid A likely differ, both the polymyxins and brevicidine form stable, long-lived interaction within the outer membrane, suggesting this could be a conserved mechanism of lipid A-binding antibiotics. However, while the three-state model suggests polymyxin B clustering is a necessary step in the activities of this class of antibiotics, our findings do not exclude other possibilities for lipid A-targeting bactericidal molecules.

The three-state model could aid in the identification and design of selective lipid A-targeting antibacterial molecules able to overcome polymyxin resistance and of potentiators able to induce outer membrane permeability. Importantly, how polymyxin B interacts with lipid A in the context of the phospholipid inner membrane, a step proposed to be necessary for cell killing[26], and with mammalian kidney cells to understand nephrotoxicity associated with polymyxin clinical use[59], remain to be determined. Overall, the three-state model provides a quantitative understanding of kinetic processes driven by the multi-faceted physicochemical properties of polymyxin B that essentially transform the LPS barrier, which has evolved to preserve cell integrity, into a catalyst that promotes accumulation of cytotoxic clusters (see Supplementary Information Section 2B). Assays that allow for rapid and accurate assessment of the interactions of molecules with the outer membrane, including SPR approaches described here, will be instrumental for understanding how antibacterials can overcome the outer membrane barrier and provide a blueprint for the design of much-needed antibiotics with activity against gram-negative bacteria.

## Methods

### Minimal Inhibitory Concentration (MIC) and antibiotic potentiation assays

Bacterial strains and plasmids are listed in SI Table S5. LB growth media was prepared according to manufacturer's instructions and supplemented with 0.2% arabinose and carbenicillin (50 μg/mL) for the strains containing pBAD24-*mcr*1 plasmid. Cultures were started by inoculating 3 ml LB with 1–2 colonies from fresh overnight plates and grown at 37 °C until in log phase. Modified minimal inhibitory concentration (MIC) assays were performed in 96-well round bottom polystyrene plates (Corning) with a final volume of 100 μl in LB supplemented with tween-80 at 0.0005%. For outer membrane permeability assays, rifampicin was added at 1.56 μM (1/4x the MIC). Polymyxin B sulfate (TCI Co. Ltd), polymyxin B nonapeptide (PMBN) (Sigma), and brevicidine (Genentech) were diluted directly in the assay plates. Bacteria were diluted in LB with tween-80 0.0005% and added to a final OD$_{600}$ of 0.0005. Growth was measured via OD$_{600}$ on a SpectraMax M5 plate reader after overnight static growth at 37 °C with humidity.

### Outer membrane vesicle (OMV) preparation

OMVs were isolated as previously described[60]. Briefly, 1 L cultures of bacteria were grown in LB overnight (approximately 16 h) at 37 °C with aeration. Cells were pelleted by centrifugation at 17,700 × *g* (15,000 rpm using a F10-6x500y rotor) for 30 min at 4 °C. Supernatants were filtered through a 0.45 μM PVDF filter (VWR) and concentrated via tangential flow filtration to a volume of approximately 50 mL. OMVs were pelleted by ultracentrifugation at 185,677 × *g* (40,000 rpm using a 45Ti rotor) for 2 h at 4 °C. OMV pellets were washed in 50 ml of OMV buffer (phosphate buffered saline [PBS, Fisher Scientific] plus 200 mM NaCl, 1 mM CaCl$_2$, and 0.5 mM MgCl$_2$) by ultracentrifugation as described above. The washed OMV pellet was

resuspended in 1.5 ml of OMV buffer and passed through a 0.45 μM PVDF syringe filter. OMV preparations were quantified using a standard Bradford protein assay. Aliquots were stored at 4 °C or at −80 °C.

To compare the composition of the OMVs to outer membranes, *E. coli ΔtolQ, E. coli ΔtolQ pmrA*$^{G53E}$, and *E. coli ΔtolQ* with pBAD24-*mcr*1 were grown as described for OMV isolation and cell pellets frozen at −20. The cell pellet was brought up in ice-cold 25 mM HEPES buffer, pH 7.4, with 1x protease inhibitor cocktail (cOmplete mini, Roche), lysed using a LVI Microfluidizer homogenizer (Microfluidics), and centrifuged for 10 min at $4000 \times g$ in a tabletop centrifuge. Supernatants were centrifuged at $250,000 \times g$ for 1 h at 4 °C (Beckman TLA 120.2). Pellets were washed in HEPES buffer plus protease inhibitors. To solubilize the inner membrane, pellets were suspended in 25 mM HEPES buffer, pH 7.4, with 2% sodium lauroyl-sarcosinate (Sigma), incubated with rotation at room temperature for 30 min, and centrifuged as before. The outer membrane protein containing pellet was suspended in OMV buffer and quantified using Bradford protein assay. 0.5 μg of each sample (prepared with BOLT LDS sample buffer and reducing agent (Invitrogen)) was separated on a 4–12% NuPAGE gel in 1x MOPS buffer (Invitrogen) and stained for 1 h with InstantBlue Protein Stain (Novus Biologicals).

## C1 chip preparation and OMV capture

Polymyxin B was covalently attached to a C1 chip (Series S Sensor Chip C1, Cytiva) using an amine coupling kit (Cytiva). Cartoon in Fig. 1A created with BioRender (biorender.com). Briefly, equal amounts of reagents N-hydroxysuccinimide and 1-ethyl-3-(3-dimethylaminopropyl) carbodiimide hydrochloride were mixed as per manufacturer's instructions and immediately added to the gold chip surface for 2 min, washed with distilled deionized water, and dried. Sufficient polymyxin B (1 mM in 1 M HEPES buffer, pH 8) was added to cover the chip surface, incubated at room temperature for 1 h, washed as before, and then incubated with 1 M ethanolamine hydrochloride-NaOH, pH 8.5, for 1 min. The chip surface was washed with water, dried as previously described, immediately loaded into a Biacore S200, and primed with running buffer (described below) to equilibrate the system. C1 chips were used for multiple runs and discarded upon removal from the Biacore S200.

SCK C1-chip SPR experiments were performed in running buffer (Dulbecco's phosphate-buffered salt solution 1x without calcium or magnesium (Fisher Scientific), pH 7.4, with 0.0005% tween-80 (Sigma) passed through a 0.2 μm filter) unless otherwise indicated. MgCl$_2$ was added as indicated in the figure legends. Inclusion of tween-80 prevented loss of polymyxin to the plastics[34] (also see Supplementary Information Methods). For experiments with brevicidine, which was suspended in DMSO, 0.125% DMSO was included in the running buffer. Analysis and compartment temperature was set to 25 or 37 °C as indicated in figure legends. OMVs were diluted from frozen stocks to approximately 20–30 μg/ml protein in the OMV buffer. Capture of OMVs was performed at a low flow rate (5 μl/ml) for 300 s over the test channel(s) followed by a 300 s stabilization period. All subsequent steps were at a flow rate of 40 μl/ml. For single cycle kinetics, two-fold dilutions were injected over the channel(s) loaded with OMVs and a reference channel without OMVs for 30 s contact and 480 s dissociation. To regenerate chips, 0.5% SDS (desorb 1, Cytiva) was injected into all channels for 60 s at 30 μl/ml twice, with an extra buffer wash and four carry-over control steps to prevent residual SDS from disrupting the following cycle.

All single-cycle kinetic SPR traces shown were double-referenced (unless indicated) within the Biacore S200 Evaluation Software 1.0, exported as a.txt file, and imported (decimated) into GraphPad Prism (version 9.3.1 for Mac, GraphPad Software, San Diego, CA). For clarity, in some cases individual outlier data points (at buffer transitions) were removed when this would not change the overall data or conclusions.

## Whole cell SPR

Bacterial strains were grown in LB or LB with 0.2% arabinose and carbenicillin (50 μg/mL) to log phase (OD$_{600}$ approximately 0.4–0.6), centrifuged for 10 min at $2500 \times g$ (3500 rpm using a tabletop centrifuge), and re-suspended in PBS to a final OD$_{600}$ of approximately 5. SPR on whole bacterial cells was performed and analyzed as described for OMVs on C1 chip, but with an additional carry-over wash prior to the capture. Regeneration of the chip after whole cell binding also required additional steps (40 μl/ml flow rate): (1) PBS supplemented with 32 mM MgCl$_2$ for 120 s, (2) 2.5 M NaCl for 30 s, (3) 0.5% SDS (Desorb 1, Cytiva) for 60 s with carry over controls between. Capturing sufficient RUs of whole bacterial cells required additional injection optimization and exhibited high experiment-to-experiment variability.

## Mechanistic studies of polymyxins binding to cells, OMVs and LPS bilayers

OMV-coated and whole cell-coated C1 chips were prepared as already described with the following changes and LPS-coated surfaces were prepared identically but with rough-LPS extracted from *E. coli* F583 (Rd mutant, Sigma #L6893). For LPS, cloudy colloid suspension (above its CMC, ~1 mg/ml) contains LPS micelles that were injected and captured. For all formats, sample buffer and running buffer were composed of Dulbecco's phosphate-buffered salt solution containing 0.0005% tween-80 (Sigma), 1 mM calcium and 0.5 mM magnesium (Fisher Scientific), pH 7.4. Analysis and compartment temperatures were both set to 37 °C. Freshly prepared OMV-coated or whole cell-coated sensing surfaces were employed for each injection of polymyxin B while LPS-coated sensing surfaces could be fully regenerated by injecting 50 mM CHAPS. Reversibly bound PMBN could be reinjected over all surfaces without requiring regeneration as it fully dissociated within a few minutes.

Serial doubling dilutions of PMBN were injected from 2.5 mM to 0.156 mM over whole cells and OMVs at 50 mL/min for 1 min (Fig. 3A, B). This was repeated over LPS-coated sensing surfaces and, in this case, serial doubling dilutions of PMBN were injected from 5 mM to 0.625 mM for 30 s at 50 mL/min, immediately prior to injection of 1 mM polymyxin B for 200 s at 50 mL/min (Fig. 3D). The PMBN injection series was immediately repeated (Fig. 3C) after a single concentration of polymyxin B (325 nM) was injected, which saturated each sensing surface (Fig. 3D).

Fresh sensing surfaces coated with whole cells and OMVs were prepared and 325 nM polymyxin B was injected at 50 mL/min for 120 s over both. This injection was repeated after a long delay, and without regeneration, in order to allow free sites to become available thereby resulting in two polymyxin B binding curves for each sensing surface (Fig. 3E, F). A fresh LPS-coated surface was prepared and polymyxin B was injected using SCK injection mode over a serial doubling dilution range from 625 nM to 39 nM. This resulted in surface saturation and the same injection series was immediately repeated resulting in the two curves shown in (Fig. 3G), where the second injection series fails to generate prolonged retention.

PMBN injections performed immediately after saturation of a sensing surface with polymyxin B reports loss in polymyxin B occupancy because they are proportional to the PMBN binding response and therefore report the increasing fraction of free LPS sites that become available through polymyxin B dissociation. Here 10 mM PMBN was injected for 30 s at 50 mL/min at regular time intervals of 850 s, over sensing surfaces coated with OMVs or whole cells and the steady-state response regions were then plotted as a function of time and fit to a dissociation model (Fig. 3H).

All coated sensing surfaces were paired to uncoated sensing surfaces providing a reference response for data analysis. Evaluation without referencing showed the C1 sensor chip surface had minimal to no non-specific binding to polymyxin B or PMBN (Supplementary Information Fig. S16).

## Modeling and statistical methods

Modeling and data analysis associated with Figs. 3 and 5 are described in detail in Supplementary Information Section 1 and 2, respectively.

## Topography SEM of OMVs on SPR chips

OMVs were loaded onto all channels of a C1 in a Biacore S200 as described above. The chip was then removed from the machine and fixed and stored in modified Karnovski's fixative (2.5% paraformaldehyde and 2% glutaraldehyde in 0.1 M cacodylate buffer, pH 7.2) for at least 24 h at 4 °C[61]. SPR chips were then washed with ultrapure water stained 4% (w/v) uranyl acetate for 15 min at room temperature. The chips were then washed again, dehydrated in a series of ascending ethanol concentrations, and finally incubated twice for 10 min at room temperature with hexamethyldisilazane (HMDS) as the final dehydration steps. The HMDS was then allowed to slowly evaporate. Air dried chips were then coated with a 2 nm thin layer of gold-palladium using a sputter coating device and examined in a Zeiss Gemini 300 Scanning electron microscope.

## Negative staining and TEM imaging

OMVs were incubated in OMV buffer (as above) plus 0.0005% tween-80 and then polymyxin B, PMBN, or an equal volume of OMV buffer, added for a final ratio of 0.5:1 polymyxin B (or PMBN) to LPS for either 1 min or 40 min. The amount of LPS in the OMV preparations was determined using fluorescently labeled LPS as described[26]. LPS was quantified by addition of 20 μL of 4 μM dansyl-polymyxin B to 20 μL of sample or dilution series of LPS in PBS with 0.004% tween-80 and 5 mM EDTA in a while polypropylene 384-well plate (Thermo-Fisher). The plate was incubated for 30 min at room-temperature before reading the fluorescence intensity on a plate reader (Ex 340 nm, Em 495 nm, cutoff 475 nm). Concentration was interpolated using a standard curve. Log-phase bacteria were diluted to $OD_{600}$ of 0.25 in 3 mL of LB supplemented with tween-80 before addition of 1 μL of 1 μM polymyxin B or PMBN and incubated statically at 37 °C. Samples were fixed at times indicated by addition of equal volume of 2x Karnovsky fix and stored at 4 °C. The suspensions were then adsorbed to the surface of formvar and carbon-coated standard TEM grids (75 mesh) for 15 min at room temperature, quickly rinsed in ultrapure water. OMVs were stained twice for 60 s with 2% aqueous uranyl acetate and bacteria were stained with uranyl-less stain (ready-to-use solution from Electron Microscopy Sciences) twice for 60 s. Excess staining solution was blotted off and grids were air dried. Imaging was with JEOL JEM-1400 transmission electron microscope (TEM) operated at 80 kV using magnification form 5000x to 100,000x. At least 20 images were taken for each sample.

## Reporting summary

Further information on research design is available in the Nature Portfolio Reporting Summary linked to this article.

## Data availability

All data supporting the findings of this study are available within the paper and the Supplementary Information. Source data are provided with this paper.

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

## Acknowledgements

We thank Jian Payandeh and Hoangdung Ho for preparations of mam-malian vesicles, Mary Kate Alexander for reagent generation, Nicholas Nickerson for constructing the ΔtolQ ΔpmrA strain, Aedan Liu for con-structing the pBAD24-mcr1 vector, and Michael Koehler for synthesis of brevicidine. We thank members of the Department of Infectious Dis-eases at Genentech for critical reading of the manuscript.

## Author contributions

M.R. performed electron microscopy. M.C.J. performed cryo-electron microscopy, and S.J.R. performed mass spectrometry. P.A.S. and S.T.R. initiated the investigation. K.R.B. and S.T.R. led the microbiology

experiments. K.R.B. and J.G.Q. performed SPR experiments. J.G.Q. conceived and quantified the model. K.R.B., J.G.Q., and S.T.R. oversaw the study and wrote and revised the manuscript. All authors analyzed the data and reviewed the manuscript.

## Competing interests

All authors, except P.A.S., are employees of Genentech, Inc., a member of the Roche Group. P.A.S. is a former employee of Genentech, Inc., and is currently an employee of Revagenix, Inc.
