## [Peer Review File · Nature Communications]

Potent activity of polymyxin B is associated with long-lived super-stoichiometric accumulation mediated by weak-affinity binding to lipid AReviewer #1 (Remarks to the Author):

This study aimed to examine the mode of action of polymyxins. However, the methodology is oversimplified by using OMVs and only biophysical methods (mainly SPR). Generally speaking, results with SPR can only conclude whether a binding event occurs, and it is risky to make further assumptions than that (especially for the uncertain/new proposed mechanism). There are several major issues in the study design and data interpretation.

MAJOR COMMENTS:

(1) Same as a recent NC paper using AFM (i.e. ref 24), the present study only used OMVs to mimic bacterial outer membrane. In lines 116-117, the authors stated "OMVs are useful surrogates for the Gram-negative outer membrane 24,41,". Whether OMVs are useful surrogates for the Gram-negative outer membrane depends on the study question. There are a number of differences between OMVs and bacterial outer membrane, such as lipid components, the proportion of multiple unmodified/modified lipid A (the authors only determined the peak area % of lipid A from OMVs [Table S1] but did not compare the results with outer membrane), as well as other components including membrane proteins. An essential requirement in this study is to demonstrate similar compositions of the OMVs and outer membrane with reliable quantitative membrane lipidomics and proteomics data. Otherwise, the data are not sufficient for the conclusion. This also applies to ref 24 in which the conclusion is not well justified.

(2) Although the SPR technology the authors have employed can provide extensive information on the binding affinity, specificity, dissociation and association rate constants for the molecular binding mechanism(s); it is not a bona fide membrane mimetic system. The ability to form model membrane assemblies, bilayers incorporating the lipids (including multiple species of lipid A/LPS) and biologics (e.g. proteins) which closely mimic the Gram-negative bacterial outer membrane is not tenable on the binding surface. The fitting of the data to a monoexponential reaction is an oversimplification of the scenario in the bacterial membrane wherein the polymyxin lipid A stoichiometry is multi-valent (ACS Infect. Dis. 2017, 3, 8, 606–619). The authors have drawn extensive and unfounded conclusions from the SPR data which are not well supported by the actual data.

Moreover, there are technical insufficiencies. Non-specific binding in SPR experiments needs to be carefully controlled, particularly with polymyxins, as it is common for these cationic analytes to bind to the SPR surface instead of exclusively to the target. This can make binding appear stronger than it actually is; and would make the sample and reference surfaces the same and only the specific binding would be seen when the sample responses were reference subtracted. The authors should demonstrate that they minimized this effect by supplementing the running buffer with additives such as dextran. When analysing positively charged polymyxins, they should block the sensor chip with ethylenediamine instead of ethanolamine after amine coupling. This will reduce the negative charge on the sensor surface and thus decrease the potential for non-specific binding.

It is important to note that the study of macromolecule-ligand interactions by SPR solely depends on the mass changes during the reaction. This becomes problematic with polymyxins which are known to remove LPS or lipids from the immobilized surface, as indicated by the time-dependent diminution in RUs. Furthermore, this indicates that the polymyxin alters the chemical environment of the surface which causes the signal to drop because of a conformational change in the immobilized surface.

(3) The authors proposed the three-state model for polymyxin binding to the outer membrane, and suggested that cPMB is essential for polymyxin antibacterial activity. However, this was only based on a mathematical model (its limitations are detailed below) and they should experimentally validate this model and the hypothesis. The authors only said that cPMB is essential but how?

(4) There are a number of issues with the modelling section. Elementary reaction kinetics was assumed for the phase separation model (lines 243–283); however, these equations might be

problematic due to incorrect reaction stoichiometric coefficients and lack of allosteric effects (e.g. hill coefficient).

- Lines 276- 278. "The resulting simulated curve (defined by Eqn. (7)) was fit to the experimental data by nonlinear regression analysis using least squares curve fitting." Typically, in curve fitting, a set of ODEs is fit to experimental observations to determine the best set of parameter estimates that describe the data (i.e. the lowest χ^2). Lines 276-278 are ambiguous and should be updated as it current read as if only equation 7 was fit to the data and all other parameters were fixed and not estimated. If this is not true, then please update lines 276-278 to improve clarity.

- Equation 3 did not account for the loss of PMB-L due to the formation of nPMB-L. Please clarify this.
- Lines 289-290: "The low standard deviation of the returned ..." SD was derived based on $n=2$. Instead of doing this, I suggest reporting the uncertainties of the parameter estimates from the model outputs.

- Lines 263-283: "Constraining the three-state model was important to ensure model fitting stability". This is a relatively complex model but with only one measured DV (i.e., RU). I am unsure whether the data are sufficient to reliably estimate all the model parameters.

- Finding good starting values is very important in non-linear regression to allow the model to converge to the global minimum. How did the authors determine the initiate values for the model parameters?

- Did the authors perform sensitivity analysis? Can the authors vary the final estimates and check model convergence and parameter estimates? Some parameters were fixed to experimentally determined values such as $k_6 = t/1/2 = \ln 2 / K_{off}$. Please perform sensitivity analysis to evaluate the impact of varying those parameters (e.g., k_6) on other parameters in the model.

- Did the authors perform any model discrimination to simplify the model? I understand the current structure is informed by the biology, but it is still important to perform model discrimination to avoid over-parameterizing the model.

- Table 2. It seems the data from different OMV concentrations were fit separately. Please elaborate and justify this. I would recommend fitting all the data from different OMV concentrations simultaneously. This would give a more reliable estimate than fitting the data separately.

- Data from wild-type and modified OMV should also be fit simultaneously. Covariate modelling (i.e. using the type of OMV as a categorical variable) should subsequently be performed to evaluate the impact of type of OMV on different model parameters.

- I would not perform summary stats for $n=2$. At least $n=3$ is needed to perform summary stats.

- Please report the SEs and the associated parametric 95% CIs for each parameter for each mode fit.

- Figures 4A and 4B.

- o Please provide figure legend for the lines.

- o Please include 95% CI for the model fits and overlay with the observed (from all independent SPR runs).

- Lines 305-307 and Table 2. "...formation of cPMB was entirely absent and this is apparent in the species component plots". Please elaborate this. How was this determined? Via modelling? Did the authors experience model convergence issues when fitting the data and subsequently decided to simplify the model by dropping the cPMB compartment? Or was k_5 estimated to a very small value and subsequently fixed to 0? The authors should provide details on how modelling was performed in the method section. Without the details, it is difficult for this reviewer to assess the M&S works.

(5) Lines 79-80: I know the statement is from ref 25; however, in ref 25 the experimental design is problematic and the data are not convincing. It is well known that the proportion of LPS in the inner membrane is much lower (compared to in the outer membrane) which should dramatically affect the damage by polymyxins even if there was such an effect.... Therefore, the assumption that polymyxins also target LPS in the inner membrane still lacks sufficient evidence.

(6) Lines 203-208: The authors compared polymyxins with brevicidine and indicated that both antibiotics shared common antibacterial mechanisms. This section is contradictory. The antibacterial mechanism of brevicidine is not clear, such as (i) how does it bind to LPS, (ii) how does it overcome

polymyxin resistance in bacteria with modified lipid A while having a “common mechanism” with polymyxins at the same time?

(7) Lines 89 and 243: the assumption of polymyxin:LPS=1 is unlikely correct. Please see several recent molecular dynamics simulations papers on polymyxins in the literature.

MINOR COMMENTS:

Lines 62-63: The mechanisms of polymyxin resistance also include LPS loss and polymyxin dependence according to recent papers.

Lines 74-75: I know ref 20 but the permeabilizing effect all depends on the proportion of unmodified lipid A in the outer membrane. If all lipid A molecules are modified, the permeabilizing effect of polymyxin-like molecules would be minimal.

Lines 170-171: It is overstated. Please see (1) above.

Figure 1B: Did the authors conduct any cryo-EM imaging to avoid the interference of sample preparation?

Line 889: Should (C) be (D)?

SI Table S1: The structures of different types of lipid A should be provided.

SI Figure S4: The structure of PMBN is wrong and Dab1 should be removed.

Reviewer #2 (Remarks to the Author):

This is a significant and well done piece of work. It has shed great light on the mechanisms by which polymyxins interact with the outer membrane of bacteria to eventually cause bacteria cell death. The work is technically well done and uses appropriate controls. The authors have done a fine job in communicating their results clearly and objectively in the context of what is known in the current literature. I found it easy to read despite what could be more challenging biophysical and mathematical components to some readers. I think the study opens up new horizons to researchers in targeting bacterial cell membranes for enhancing molecules or new chemical entities to destroy bacterial cell membranes.

I only have some minor points for clarification/improvement.

The EM date in Figure S1 is decent but the significance of changes induced by polymyxin are a little murky. I suggest a semiquantitative approach to presenting data from images. Perhaps, changes in OMVs per area or tubules per OMVs, etc. can be considered. Some changes are discerned from the images by eye but how prominent they are is harder to interpret.

How did the authors minimize refractive index change from buffer compatibility with increasing concentrations injected. Was any background refractive change apparent?

Reviewer #3 (Remarks to the Author):

The authors describe a study investigating how a class of antibiotic (polymyxins) interacts with a molecule called lipid A, which is found in the protective outer layer of several gram-negative bacteria. Cationic polymyxins are known to bind to negatively charged lipid A, resulting in a conformational change that ultimately leads to pores forming in the membrane and the killing of the cell. However much is still unknown about this process, and in order to understand it better, the researchers used surface plasmon resonance (SPR) to study the stages of interaction between polymyxins and lipid A. Second, they develop a mechanistic model to explain how transient binding of polymyxins to LPS facilitates incorporation into the bacteria, promoting phase separation and the emergence of high density clusters. They propose a mechanism by which killing of cells would be consistent with their SPR data, model, and previously reported investigations. This mechanism could be a useful insight for the development of other antibiotics targeting similar molecules, and could be an insight into how cells develop resistance to polymyxins by failing to get killed. The premise of the article is potentially an important contribution to the high priority field of antibiotics and antibiotic resistance. However, I have some concerns which I feel should be addressed before proceeding.

Major

The authors propose a 6 parameter model that is then fit to the SPR data. The high number of parameters runs the risk of over-fitting. Usually simpler models (for example a 2 state model with 4 parameters in this case) should be tested first and found to be insufficient to explain the data before moving to more parameters. The authors should provide adequate quantitative justification for their model complexity, e.g. by showing why a simpler model would not suffice.

The authors do not provide any form of systematic cross validation study for their model, i.e. to test its predictive capability in contexts other than the data used to parameterize the model. For example they could have used leave-one-out cross-validation or k-fold cross-validation, i.e. methods where the model is parameterized with a subset of the available data and then tested for validity against the remaining data.

The parameters determined from fitting the model to SPR data shown in Table 2 are determined with the barest minimum of statistical rigor, with $n=2$, and there is no discussion of the type of replicates used – are these just technical replicates done on the same sample? Or, if I am not mistaken – it seems like 2 *different* concentrations were used, then averaged, and then this is called $n=2$? If this is the case, then that does not make any sense! In table 2 for the second-to-last item KD cPMB, it looks as though the points 0.186 and 0.202 are both way outside the reported mean \pm standard deviation (0.194 \pm 0.01). Or are the replicates actually distinct experimental samples that were prepared independently? In any case, there appears to be insufficient transparency about the quality of statistical sampling, reproducibility, or shape of the distributions being investigated.

The large number of model parameters also creates the risk for degenerate solutions – where multiple sets of very different values of the rate constants could achieve equally good fits. The authors should provide evidence that their model is sufficiently constrained such that this is not the case – either by fitting to a subset of the data for example and showing that parameter values do not change drastically from their current form. Or by conducting more experimental replicates and showing that the distribution of parameter values is well-behaved - i.e. normally distributed around a single value.

One of the major claims in this paper is that the modified version of polymyxin B (PMBN) lacks the long term membrane accumulation effect of polymyxin B. While the authors show the accumulation effect of polymyxin B on whole E. coli cells, and unless I am mistaken, they do not show how E. coli cells respond in the presence of PMBN. This seems like an important experiment to include to support the primary claim of this paper and the validity of OMVs as a model system in this context, especially considering how artificial the model system is, using tween-80 for example which is likely to affect lipid properties.

The authors claim that TEM imaging of outer membrane vesicles shows how polymyxin causes vesiculation and tubulation. However they use fixation/drying (as opposed to cryo-EM for example), so isn't this an approach that is likely to introduce artifacts? Likewise for the SEM image in Fig 1C – it is hard to say whether this is representative of intact surface coupling of aqueous-liquid-packed spherical vesicles given that these samples must be deposited, dried, and sputter coated with a layer of metal before imaging. Are there other articles to support these dry methods as a valid technique for characterizing lipid vesicles? References 25 and 62 cited for these steps respectively do not seem to show this.

In methods, the authors state: "Sensorgrams could not be normalized for bound ligand (i.e., the OMVs on the chip surface) using a calculated 'Rmax', and, as such, analyzed traces, including those in the figures, were selected to have similar levels of loaded OMVs." Does this mean that the authors manually selected which runs to include? How many runs were discarded in this process? What did they look like? Such a practice is going to be prone to human selection bias – the proper way to do with would be to choose an objective formal selection criterion and state it along with the actual variation in bound ligand between samples that were included.

Minor

The lettering order in Fig 2 is confusing, with order going columnwise and then rowwise in the same panel.

Fig 1 D and E are a plot of the same data, and it is unnecessary to show it twice. The annotation could be added to the first plot to communicate the same effect, and it would be less misleading as it currently looks at first glance like two different experiments.

Polymyxins are known for exhibiting nephrotoxicity, and the mechanism for this toxicity is thought to involve binding to phospholipids in the cell membrane. Could the authors comment on any similarities or differences between human versus bacterial cell toxicity and any insight their model might or might not provide here – for example regarding differences or similarities in membrane organization?

We would like to thank our reviewers for their thoughtful and thorough reading of our manuscript, “Potent bactericidal activity of polymyxin B is enabled by low-affinity lipid A-binding coupled with phase separation” (revised title), and for providing constructive and insightful comments. We have experimentally and computationally addressed these comments, improving the manuscript significantly and are confident that it will be of high interest to the field. We have provided point-by-point responses to each comment below. Moreover, motivated by the reviewers’ comments and in an effort to provide as much clarity to our model as possible, we have made the following significant revisions:

- We performed cryo-EM studies with outer membrane vesicles (OMVs) in the presence of polymyxin B to confirm our previous TEM observations. These results demonstrate that OMVs are perturbed by polymyxin B but not the non-bactericidal polymyxin B nonapeptide (PMBN). Although it still remains to be determined what role these physical changes have in the molecular mechanism of polymyxin’s bactericidal activity, we provide these observations as a means to support the proposal that OMVs can be used as proxy tools for understanding specific features of polymyxin’s interaction with the outer membrane barrier. Additional details are described below and in the revised text
- We have provided additional support throughout for the use of OMVs in our approach, and also, importantly, have carried out additional confirmatory experiments with whole bacterial cells as well as purified lipid A. Our findings with all three formats are consistent with each other and with our more complete mechanistic model, which provides new insight into polymyxin’s mechanism and activity as highlighted in our revised discussion.
- Although our overall conclusions remain essentially unchanged, we have provided a much more complete analysis and detailed mechanistic model for polymyxin binding. A detailed explanation of our model, its derivation and associated methods is provided in the supplementary information section in full, with sufficient detail to enable immediate application by others in the field.
- A brief summary in the main text now describes our systematic approach to deriving the the model which has been upgraded to include a transition state.
- The introduction of Kintek Explorer for model discovery and validation was key to this progress as this tool is specifically designed for this purpose and incorporates advanced numerical and statistical methods for robust model development and validation. Specifically, re-working of our model was enabled by the ability to readily test models using 2D FitSpace analysis (Johnson et al. (2009) *Analytical Biochemistry*) which provides a robust way to ensure that all parameters are well-constrained within acceptable bounds over broad ranges in parameter values while evaluating whether the fit is unique.
- We also developed new SPR analysis formats and a finite element analysis-based simulation to produce data sets tailored to establish the fundamental tenets of our model.
- While elements of other published work in the field appear consistent with our findings we acknowledge that there is insufficient evidence to support a full physical understanding. Therefore we have re-written the discussion section to focus on our current results and less on the possible implications for the further development of a full physical understanding.

We have been able to advance this work significantly beyond our original manuscript and we now submit a revised manuscript for your consideration for publication in *Nature Communications*.

Reviewer #1 (Remarks to the Author):

This study aimed to examine the mode of action of polymyxins. However, the methodology is oversimplified by using OMVs and only biophysical methods (mainly SPR). Generally speaking, results with SPR can only conclude whether a binding event occurs, and it is risky to make further assumptions than that (especially for the uncertain/new proposed mechanism). There are several major issues in the study design and data interpretation.

Response 1: We thank the reviewer for their thoughts on our work. That polymyxins bind lipid A is universally accepted, however even a cursory examination of the literature reveals that the nature of this interaction is ill-defined. The involvement of lipophilic interactions in retaining polymyxin B at the outer membrane is well known and here SPR reveals how such interactions manifest in producing extremely high and long-lived polymyxin B loading via phase separation. We respectfully disagree that SPR can only inform on whether binding occurs (as the reviewer indicates below in point #2, SPR data is rich with information) and this approach has been used extensively to inform on mechanisms in diverse applications. While SPR has been used extensively for evaluating affinity binding it can also be applied to phase transitions (for example, see Aibara et al. (2016) *Journal of Physical Chemistry C*) because a change in phase state will be accompanied by a change in stability relative to the previous state which is observable in the kinetic curvature measured by SPR. The true mechanism is therefore already encoded in the curves and the objective of the iterative modeling process is to discover this mechanism. While we agree that our approach and interpretations can only go so far, we are confident that we have contributed a significant and novel insight into the mechanism of polymyxins. We propose that our kinetic model not only rationalizes our data, but is consistent with other work and explains multiple activities of this important antibiotic class. We detail both these points and the use of OMVs in our study in the responses below. We hope that our responses (and accompanying changes to the manuscript including a complete reworking of the computational modeling sections) allay some of the reviewer's concerns.

MAJOR

COMMENTS:

(1) Same as a recent NC paper using AFM (i.e. ref 24), the present study only used OMVs to mimic bacterial outer membrane. In lines 116-117, the authors stated "OMVs are useful surrogates for the Gram-negative outer membrane 24,41,". Whether OMVs are useful surrogates for the Gram-negative outer membrane depends on the study question. There are a number of differences between OMVs and bacterial outer membrane, such as lipid components, the proportion of multiple unmodified/modified lipid A (the authors only determined the peak area % of lipid A from OMVs [Table S1] but did not compare the results with outer membrane), as well as other components including membrane proteins. An essential requirement in this study is to demonstrate similar compositions of the OMVs and outer membrane with reliable quantitative membrane lipidomics and proteomics data. Otherwise, the data are not sufficient for the conclusion. This also applies to ref 24 in which the conclusion is not well justified.

Response 2: We appreciate the reviewer's questions about the use of OMVs, however we offer that, for the purposes of our study, they are powerful surrogates for studying the interactions between polymyxin and lipid A and note the same binding profiles were shown using whole cell bacteria (revised **Fig. 2**). We have provided additional literature precedent and OMV analyses in our revision (in both the introduction and results sections) to better define what these vesicles can (and cannot) capture about the outer membrane environment as it relates to polymyxin binding. We are now careful to contextualize their application in our current study. Critically, we now confirm our results using whole *E. coli* cells. In brief, our SPR observations and model fits for both OMVs and whole cells are well-defined and in complete agreement (revised **Fig. 2** and **3**). We have also added additional data with purified LPS (revised **Fig. 3**) that demonstrate the observed

kinetics depend solely on the interactions with this target. Thus, we feel we can conclude that OMVs capture all of the relevant interactions required for the binding of polymyxin B that we observe by SPR. We also more directly outline why the use of OMVs is advantageous in the study of polymyxin binding by SPR. As an aside, OMVs in our approach are distinct from previous uses of OMVs (specifically in Maniglu et al. (2022) *Nature Communications* where the OMVs are flattened on a mica surface). Given the limitations of cells, isolated LPS, or other membrane mimetics, we propose that our studies establish OMVs as a tractable tool to study aspects of polymyxin interactions with the outer membrane.

(2) Although the SPR technology the authors have employed can provide extensive information on the binding affinity, specificity, dissociation and association rate constants for the molecular binding mechanism(s); it is not a bona fide membrane mimetic system. The ability to form model membrane assemblies, bilayers incorporating the lipids (including multiple species of lipid A/LPS) and biologics (e.g. proteins) which closely mimic the Gram-negative bacterial outer membrane is not tenable on the binding surface. The fitting of the data to a monoexponential reaction is an oversimplification of the scenario in the bacterial membrane wherein the polymyxin lipid A stoichiometry is multi-valent (*ACS Infect. Dis.* 2017, 3, 8, 606–619). The authors have drawn extensive and unfounded conclusions from the SPR data which are not well supported by the actual data.

Response 3: We hope that our rationale, now more clearly stated, addresses the reviewer's concerns. As we pointed out above, we have provided evidence that the OMVs (which contain biological complexity and do not need to be synthetically built on a surface) do provide a complete picture of binding as seen on whole cells as measured by SPR, which are now given a more prominent place in the revision, and purified LPS.

We note that we employ a three-state model that also includes mass transport to the surface, so the model is composed of multiple exponential phases, not one. As is always the case with binding kinetics, it is the slowest kinetic process that defines the observable kinetic curvature and, therefore, not all component exponentials are apparent at all times. Indeed, this depends on many factors, especially the concentration regimes of each reactant and species formed at any given time. Each state has quite different kinetics so resolving the issue of rank-deficient superimposed exponential processes is not problematic.

Regarding the point about stoichiometry, we note, now more directly, that while the model follows mass action chemical kinetics it cannot be regarded as a microscopically precise mechanism with the exact stoichiometry given the absence of supporting structural evidence. However, the average binding stoichiometry over the three states can be estimated from the SPR binding (**SI Section 1G**) but this is not taken to specify the 'exact binding valency' of the wide ensemble of microscopic polymyxin B-lipid A interaction states. The model does reveal the relative fraction of polymyxin B associated with each state and we have used the words stoichiometrically and stoichiometric to mean the average binding ratio which is conventional in the SPR field.

Finally, regarding the conclusions drawn from our SPR data, we would like to emphasize that the three-state model was derived from the information richness of the SPR data itself and not from any preconceived expectations. Assigning what each species in the model might physically represent was arrived at from the laws of mass action applied to experimental data and the many controls we performed. Having arrived at the model we then took the opportunity to comment on what the model might imply microscopically as this speaks to the significance of the model in terms of formulating future testable hypotheses. While we are confident that the functionally three-state mechanism is accurate, we emphasize throughout that our suggested structural depiction

of this mechanism (**Fig. 4**) is not considered structurally proven and will likely require refinement as such details are resolved in the field.

Moreover, there are technical insufficiencies. Non-specific binding in SPR experiments needs to be carefully controlled, particularly with polymyxins, as it is common for these cationic analytes to bind to the SPR surface instead of exclusively to the target. This can make binding appear stronger than it actually is; and would make the sample and reference surfaces the same and only the specific binding would be seen when the sample responses were reference subtracted. The authors should demonstrate that they minimized this effect by supplementing the running buffer with additives such as dextran.

Response 4: As shown in **SI Fig. S4** some chip surfaces can be problematic; this is shown by the high binding levels to the standard vesicle binding chip surface (LP) which has high binding to the blank (no OMV) surface making it impossible to resolve lower concentrations. The suggestion to use dextran may come from an assumption that we are using a hydrogel-based sensor chip, such as a CM5 chip, but we are using C1 sensor chips which have no dextran, or lipophilic groups, and produce minimal non-specific binding on exposure to polymyxin B, which was an important criteria for chip selection. Even without reference curve subtraction the non-specific binding response was minimal as shown in SI Figure S16. Briefly, 1 μM polymyxin B was injected over the sensing regions of a C1 sensor chip where LPS was coated onto one channel and a reference channel without coated LPS was employed to estimate non-specific binding. Binding was undetectable (-6 RU) to the reference surfaces (no LPS on surface) while >400 RU specific binding of polymyxin B was measured to the adjacent LPS-containing surface. This was also repeated for injection of 10 μM PMBN and non-specific binding was again undetectable (i.e., -0.4 RU) to the reference surfaces (no LPS on surface) while >368 RU specific binding of polymyxin B was measured to the adjacent LPS-containing surface (**SI Fig. S16** in revised work).

When analyzing positively charged polymyxins, they should block the sensor chip with ethylenediamine instead of ethanolamine after amine coupling. This will reduce the negative charge on the sensor surface and thus decrease the potential for non-specific binding.

Response 5: This comment is addressed in Response 4, above.

It is important to note that the study of macromolecule-ligand interactions by SPR solely depends on the mass changes during the reaction. This becomes problematic with polymyxins which are known to remove LPS or lipids from the immobilized surface, as indicated by the time-dependent diminution in RUs. Furthermore, this indicates that the polymyxin alters the chemical environment of the surface which causes the signal to drop because of a conformational change in the immobilized surface.

Response 6: We agree that appreciable removal of LPS or lipids would result in a negative response curvature that would then superimpose on the response to produce extremely complex binding curves but these were not observed. This extreme baseline stability we observe (revised **Fig. 3E** and **3F**) with whole cells and OMVs over many hours conclusively implies that significant mass loss does not occur. Bulky OMVs and cells are captured at a solid phase within the diffusion boundary where flow is near zero so any blebs, or fibrils, that might emerge from an OMV, or cell, would largely be retained on our SPR surface as all are extremely large, low-diffusion particles. The cells, and certainly the OMVs, are not actively growing and these complex processes of blebbing and fibril formation may require active metabolic turnover of other constituents.

(3) The authors proposed the three-state model for polymyxin binding to the outer membrane, and suggested that cPMB is essential for polymyxin antibacterial activity. However, this was only based on a mathematical model (its limitations are detailed below) and they should experimentally validate this model and the hypothesis. The authors only said that cPMB is essential but how?

Response 7: In the revised manuscript, we have added **Fig. 3**, which provides a detailed account of how the three-state model was derived from multiple, different experiments where each experiment was tailored to address specific questions. We provide full details of these experiments in our expanded supplemental SI Sections (see **SI Section 1B-1G**) and a full account of our interpretation. This also includes finite element analysis-based modeling (**SI Fig. S13**) to validate our treatment of mass transport effects.

Polymyxin B loading levels, equivalent to complete monolayer coverage of the entire outer membrane, have been confirmed. Given these findings it is tempting to speculate that such a mass of polymyxin B aggregates at the outer membrane could cause catastrophic disruption of outer membrane integrity but also ultimately affect the inner membrane due to its proximity and the probable scale of aggregated polymyxin B clusters. However, we acknowledge that conclusively demonstrating that the extreme polymyxin B loading levels cause cell killing is a very important next step. However, we do feel that the current work, which focuses on the functional mechanism of polymyxin B binding and long-lived retention, is of high value to the field in itself and merits publication. Indeed, publishing our current work would enable the research community to participate in testing the cell killing hypotheses and accelerate elucidation of this complex mode-of-action.

(4) There are a number of issues with the modeling section. Elementary reaction kinetics was assumed for the phase separation model (lines 243-283); however, these equations might be problematic due to incorrect reaction stoichiometric coefficients and lack of allosteric effects (e.g. hill coefficient).

Response 8: In the revised manuscript, we introduced Kintek modeling, which is designed specifically for reaction modeling, to further optimize our model and we still arrived at a three-state clustering model. Optimization did result in addition of a transition state (which remains at insignificant relative fractions (<0.1%)) that facilitates polymyxin B clustering via simple self-self interactions. We have added no additional term beyond what is specified by the reaction scheme in the revised **Fig. 4** other than an SPR sensitivity coefficient. This sensitivity coefficient is associated with an increase in the SPR sensitivity due to phase change and is not related to cooperative effects.

We can certainly understand why the Hill coefficient comes to mind and other membrane-related cooperativity models (for example, Monod et al. (1965) *Journal of Molecular Biology* and Changeux et al. (1967) *PNAS*). Clustering processes usually show profound cooperativity (for example, Dasgupta et al. (2020) *eLife*), but our polymyxin B binding curves appear well fit to our model without cooperativity. Sigmoidal contributions due to cooperativity would cause significant deviations from exponential decay but this was not observed.

However, we do observe an upward drift in the response curves when far from saturation and this is well resolved during the dissociation phases (revised **Fig. 5A**). This cannot be mass accumulation since only buffer is flowing over the sensing channels between injection phases. It is known that polymyxin B binding is associated with changes in the phase state of LPS (Paracini et al. (2018) *PNAS*) and our work here supports clustering of polymyxin B and both these phase changes will cause some level of mass redistribution throughout the sensing surface. Therefore

we would expect that mass redistribution into denser phases will manifest as an increase in SPR sensitivity due to a higher refractive index increment of the denser phase and mass redistribution towards the sensing surface where sensitivity is higher (see Jung et al. (1998) *Langmuir* and Dejeu et al. (2018) *Journal of Physical Chemistry C*). This increasing sensitivity therefore applies only to the cluster state and will be most apparent during the dissociation phase, while the system is not at steady state and not close to saturation. Therefore a sensitivity coefficient (m) associated with polymyxin B clustering (cPMB) was included in the cumulative SPR response (see *Eqn (1)*). Our model may explain the absence of cooperativity since polymyxin B clustering is stoichiometrically limited by the availability of membrane insertion sites (n), which become available at a fixed rate towards a maximum.

- Lines 276- 278. “The resulting simulated curve (defined by Eqn. (7)) was fit to the experimental data by nonlinear regression analysis using least squares curve fitting.” Typically, in curve fitting, a set of ODEs is fit to experimental observations to determine the best set of parameter estimates that describe the data (i.e. the lowest χ^2). Lines 276-278 are ambiguous and should be updated as it current read as if only equation 7 was fit to the data and all other parameters were fixed and not estimated. If this is not true, then please update lines 276-278 to improve clarity.

Response 9: We had used informal notation and have rightly adopted the correct mathematical notation in the revised manuscript and SI Sections.

- Equation 3 did not account for the loss of PMB-L due to the formation of nPMB-L. Please clarify this.

Response 10: The model from the first submission was intended as an approximate semi-empirical model. However, with addition of many orthogonal experiments and state-of-the-art modeling software (specifically, Kintek) we have been able to greatly improve our ability to iteratively optimize the model and have arrived at a model that now follows mass action chemical kinetics. The rate of interconversion of each physical state and the relative fraction of polymyxin B in each state are defined. However, the volume averaged SPR response does not permit discrete spatial resolution thereby preventing the number and size of clusters be estimated. We are confident that this functional model does reveal the fundamental processes that drive polymyxin B loading but it will likely require refinement as a full physical understanding becomes available.

- Lines 289-290: “The low standard deviation of the returned ...” SD was derived based on $n=2$. Instead of doing this, I suggest reporting the uncertainties of the parameter estimates from the model outputs.

Response 11: In our revised manuscript, we have implemented full global analysis over three multi-dose titration curves, containing 13 injection segments, providing SE, CL for each parameter with a rigorous analysis of parameter bounding and uniqueness (2D Fitspace).

- Lines 263-283: “Constraining the three-state model was important to ensure model fitting stability”. This is a relatively complex model but with only one measured DV (i.e., RU). I am unsure whether the data are sufficient to reliably estimate all the model parameters.

Response 12: As mentioned above, we have implemented full global analysis over three multi-dose titration curves, containing 13 injection segments, providing SE, CL for each parameter with a rigorous analysis of parameter bounding and uniqueness (2D Fitspace). However, we have also introduced more constraints to limit the number of parameters that must be fit. For example, we

model binding of polymyxin B to LPS using the boundary layer model (see **SI Section 1A - Modeling**) which depends only on a mass transport constant (k_t) and affinity constant (obtained at steady-state in a separate experiment). k_t , not binding kinetics, is driving the observed curvature associated with formation for state 1 and this constraint links this curvature to the LPS concentration which in turn allows k_t to become a well-fit bounded parameter. Systematic simplification of the model found that forward and reverse rates for membrane insertion (k_3/k_4) appear twice thereby reducing the number of fitted parameters despite adding a transition state which improved the overall fit quality.

- Finding good starting values is very important in non-linear regression to allow the model to converge to the global minimum. How did the authors determine the initiate values for the model parameters?

Response 13: The boundary layer model (*Eqn (S3)*) provided good estimates of K_{D1} for reversible binding of PMBN to LPS (**Fig. 3A** and **3B**). Briefly, *Eqn (S3)* was fit to this data using Biaevaluation (leading SPR- analysis software) where the initial guess for K_{D1} was obtained by first fitting an affinity model to the steady-state regions of these same data sets. k_t was calculated from theory (Goldstein et al. (1999) *Journal of Molecular Recognition*) and a global fit returned values for K_{D1} and k_t that were then used as starting values for the three-state model fit of PMBN binding to LPS (revised **Fig. 5**). Kinetic constants for state 3 tended to zero while the other fitted parameters remained resolved. Thus state 3 was eliminated from the model, effectively by the curve fitting minimization algorithm, resulting in the fit in revised **Fig. 5B**. The three-state model was then fit to the polymyxin B data where initial parameters for k_t , K_{D1} , k_3 , and k_4 were taken from the PMBN fit. The simulation did not resemble the data set and therefore the initial values for the added kinetic rate constants associated with the transition state and state 3 were iterated manually. This was performed using Kintek's dynamic simulation function where one can drag the value of any parameter over a wide range of values in just a few seconds and the simulation, which is overlaid on the actual curves to be fit, responds in real time (no apparent delay) such that the observed curvature and scaling of the binding curves to be fit is roughly reproduced in the simulated curves and requires just a few minutes of manual iteration. With these initial values a global fit showed some instability due to the higher number of parameters being estimated. Therefore K_{D1} was held constant at the value taken from the independent affinity fit in **SI Fig. S14**. The fit was repeated and it was noted that k_3 and k_4 were repeating as k_6 and k_{10} , respectively, and this removed another two rate constants leaving just three binding constants and k_t to be estimated. A fit with these constraints produced a high quality fit as shown in (**Fig. 5A**) where initial values were found by manual dynamic simulation and repeating this fitting process multiple times had no significant effect on the results.

- Did the authors perform sensitivity analysis? Can the authors vary the final estimates and check model convergence and parameter estimates? Some parameters were fixed to experimentally determined values such as $k_6 = t/1/2 = \ln 2 / K_{off}$. Please perform sensitivity analysis to evaluate the impact of varying those parameters (e.g., k_6) on other parameters in the model.

Response 14: There is a vast literature on statistical and numerical methods associated with model validation and opinions vary as to the most robust approaches. However, the methods applied in Kintek Explorer, and applied here, represent state-of-the-art in mechanistic model fitting and have been developed and applied by world leaders in enzymology (Johnson et al. (2009) *Analytical Biochemistry*). The model from the initial submission was an approximate semi-empirical model that was incompletely optimized. However, we have evolved the model considerably and this was greatly facilitated by using Kintek Explorer for fitting the data in the

revised **Fig. 5**. In particular, this includes a 2D fitspace analysis that comprehensively establishes the quality of the fit.

- Did the authors perform any model discrimination to simplify the model? I understand the current structure is informed by the biology, but it is still important to perform model discrimination to avoid over-parameterizing the model.

Response 15: As described above (Response 7), the model was arrived at iteratively without bias in terms of how polymyxin B might bind. However, we had performed many SPR experiments not included in the original submission that helped constrain the model and we have now included representative data sets in **Fig. 3** and associated data in **SI Sections 1B-1E**. Our explanation of iteratively optimized the model and its constraints are also described above (Responses 7, 8, 10, 12, and 13).

- Table 2. It seems the data from different OMV concentrations were fit separately. Please elaborate and justify this. I would recommend fitting all the data from different OMV concentrations simultaneously. This would give a more reliable estimate than fitting the data separately.

Response 16: We have implemented full global analysis over three multi-dose titration curves, containing 13 injection segments, providing SE, CL for each parameter with a rigorous analysis of parameter bounding and uniqueness (2D Fitspace)

- Data from wild-type and modified OMV should also be fit simultaneously. Covariate modelling (i.e. using the type of OMV as a categorical variable) should subsequently be performed to evaluate the impact of type of OMV on different model parameters.

Response 17: Also see our Response 14 above. We can understand this request in view of the limitations of our previous modeling but the re-optimized model has performed well as indicated unambiguously by the fit validation tests (1D/2D FitSpace, SE, CI, χ^2). The main goal of our modeling was to allow a direct comparison of the rate constants for a quantitative comparison of how polymyxin B binding to mutant-OMVs differs from binding to wt-OMVs and this has been achieved rigorously in our revision. We fit the full re-optimized three-state model to the resistant-OMV data set and it spontaneously returned near zero rate constants for state 3, effectively eliminating this state from the fit, while returning reasonable values for the other fitted parameters. **Fig. 5** and **Table 2** clearly show that binding of polymyxin B to resistant-OMVs is dominated by transient interactions with LPS and slow formation of nucleates but unlike with wt-OMVs these nucleates do not lead to formation of clusters. This mechanism is consistent with the ability of lipid A modifications that lead to polymyxin resistance to stabilize the LPS affinity network (e.g., Khondker et al. (2019) *Communications Biology*), even in the absence of metal ions, thus preventing membrane stretching which appears essential to polymyxin B clustering.

- I would not perform summary stats for n=2. At least n=3 is needed to perform summary stats.

Response 18: In the revised manuscript, we have implemented full global analysis over three multi-dose titration curves, containing 13 injection segments, providing SE, CL for each parameter with a rigorous analysis of parameter bounding and uniqueness (2D Fitspace)

- Please report the SEs and the associated parametric 95% CIs for each parameter for each mode fit.

Response 19: The 2D fitspace provided informs on the confidence intervals over ranges in all pairs of fitted parameters which allows covariance to be included providing an extremely robust test of the model fit. **Table 2** contains the SE of the fit for each parameter and the CI limits.

- Figures 4A and 4B.

- o Please provide figure legend for the lines.

- o Please include 95%CI for the model fits and overlay with the observed (from all independent SPR runs).

Response 20: This has been addressed with addition and revision of **Fig. 5** and **Table 2**. Additionally, please see our Response 14 above.

- Lines 305-307 and Table 2. "...formation of cPMB was entirely absent and this is apparent in the species component plots". Please elaborate this. How was this determined? Via modelling? Did the authors experience model convergence issues when fitting the data and subsequently decided to simplify the model by dropping the cPMB compartment? Or was k_5 estimated to a very small value and subsequently fixed to 0? The authors should provide details on how modelling was performed in the method section. Without the details, it is difficult for this reviewer to assess the M&S works.

Response 21: The species component plot was generated by simply separating the components of *Eqn (1)* for the fitted curves. Therefore they are an element of the reported fit data set in **Fig. 5** and **Table 2**. We have now provided details of the modeling procedure in the main manuscript and in SI Section 1. We provided rationale for the reduction of the three-state model to just two states for polymyxin B binding to resistant-OMV is given above (Response 17). Briefly, we fit the full re-optimized three-state model to the resistant-OMV data set and it spontaneously returned near zero rate constants for state 3, effectively eliminating this state from the fit. **Fig. 5** and **Table 2** clearly show that binding of polymyxin B to resistant-OMVs is dominated by transient interactions with LPS and slow formation of nucleates but unlike wt-OMVs these nucleates do not lead to formation of clusters. This mechanism is consistent with the ability of polymyxin-resistant modified-lipid A to stabilize the LPS affinity network (e.g., Khondker et al. (2019) *Communications Biology*), even in the absence of metal ions, preventing membrane stretching which appears essential to polymyxin B clustering.

(5) Lines 79-80: I know the statement is from ref 25; however, in ref 25 the experimental design is problematic and the data are not convincing. It is well known that the proportion of LPS in the inner membrane is much lower (compared to in the outer membrane) which should dramatically affect the damage by polymyxins even if there was such an effect.... Therefore, the assumption that polymyxins also target LPS in the inner membrane still lacks sufficient evidence.

Response 22: We have been able to reproduce the experiments described in this reference (Sabnis et al. (2021) *eLife*) in multiple gram-negative species and though the interpretation of the experiments in that manuscript as well as other data are still emerging, we find that this model (in whole if not details) is parsimonious with much data in the field. Our kinetic model pertains to binding and retention of polymyxins but does not provide a physical understanding to enable conclusions related to microscopic structure or precise localization of the bound polymyxins. Such details are still emerging in the field but we felt it was reasonable to provide possible interpretations of our data extended to physical states and processes but these conjectures were intended to provoke productive discussion in the field and were not intended to be taken as scientific findings. To avoid such misunderstandings we have dramatically reduced the discussion to focus on our proposed model, though we do still mention this intriguing possibility of interaction

at the inner membrane especially because we, and others, have been limited to investigating outer membrane interactions.

(6) Lines 203-208: The authors compared polymyxins with brevicidine and indicated that both antibiotics shared common antibacterial mechanisms. This section is contradictory. The antibacterial mechanism of brevicidine is not clear, such as (i) how does it bind to LPS, (ii) how does it overcome polymyxin resistance in bacteria with modified lipid A while having a “common mechanism” with polymyxins at the same time?

Response 23: While this remains to be proven there is evidence to suggest that they do share a common membrane accumulation mechanism. The binding curves for brevicidine show an association phase profile with a transient component that dissociates more rapidly and one, or more, other components that dissociate far more slowly. This is qualitatively similar to the polymyxin B binding profile for binding to wt-OMVs suggesting that there is a possibility that it follows the three-state model, where the values of the rate constants might be quite different. For example, the ability of brevicidine to bind to resistant-LPS may follow a similar electrostatically dominant state 1 (promoting K_{D1}) with states 2 and 3 accumulating over resistant-LPS via a higher propensity to phase separate through lipophilic interactions. These lipophilic interactions would need to generate sufficient energy to oppose the stability of the resistant-LPS affinity network for the rate of formation of state 3 to become significant (i.e., $k_5 \gg 0$). The associated clusters would accumulate through stretching of the membrane. Given these assumptions, polymyxin B may not overcome resistance because it has a comparatively lower phase separation potential relative to brevicidine and is prevented from forming clusters on exposure to cells possessing resistant, modified lipid A. In other words, the three-state binding model simplifies to a two-state model when the LPS affinity network is highly stable as observed for polymyxin B over wt-OMV and resistant OMVs. Going beyond the framework of our binding model we hold that these processes may explain differences in cell killing and this would imply they would share the same cell killing mechanism but this remains to be established and will be the subject of future research.

(7) Lines 89 and 243: the assumption of polymyxin:LPS=1 is unlikely correct. Please see several recent molecular dynamics simulations papers on polymyxins in the literature.

Response 24: We agree that weaker electrostatic binding that does not conform to an average stoichiometry of 1:1 complex is likely at higher concentrations where the excess of polymyxin B allows multiple polymyxin B molecules to interact with a single LPS molecule. However, we find an average binding stoichiometry close to 1:1 in agreement with the published NMR structure which suggests that this is the favored state within the ensemble of allowable states. We include this calculation in the new SI Section 1G.

Calculation of Binding Stoichiometry

Pure LPS vesicles bound to a sensing surface were used to generate the PMBN and PMB binding data shown in **Fig. 3A-3D**. The average stoichiometry of polymyxin B occupancy by SPR can be estimated from these experiment as follows:

- Total polymyxin B bound at saturation = 475 RU
- Total LPS coated = 1500 RU
- MW of rh-LPS = ~3500 Da, MW of polymyxin B = 1203 Da, MW ratio = 3500 Da/1203 Da = 2.91
- Thus, the expected polymyxin B binding response assuming 1:1 stoichiometry = 1500 RU/2.91 = 515 RU
- Therefore, binding stoichiometry = 475 RU/515 RU = 0.92 (or polymyxin B:LPS = 0.92:1)

This calculation was also performed for PMBN binding to the same surface.

- Total PMBN bound at saturation = 414 RU
- Total LPS coated = 1500 RU
- MW of rh-LPS = ~3500 Da, MW of PMBN = 963 Da, MW ratio = 3500 Da/963 Da = 3.63
- Thus, the expected PMBN binding response assuming 1:1 stoichiometry = 1500 RU/3.63 = 413 RU
- Therefore, binding stoichiometry = 414 RU/413 RU = 1.00 (or PMBN:LPS = 1: 1)

MINOR COMMENTS:

Lines 62-63: The mechanisms of polymyxin resistance also include LPS loss and polymyxin dependence according to recent papers.

Response 25: We thank the reviewer for raising this point. We have included additional mechanisms of resistance in the revised introduction for completeness.

Lines 74-75: I know ref 20 but the permeabilizing effect all depends on the proportion of unmodified lipid A in the outer membrane. If all lipid A molecules are modified, the permeabilizing effect of polymyxin-like molecules would be minimal.

Response 26: We respectfully disagree. The results observed in this reference (MacNair et al. (2018) *Nature Communications*) have been reproduced and in the strains used in our study, we observed permeabilization was induced when over 90% lipid A was modified. We have not been able to find or generate data to suggest that the amount of modified lipid A will affect the degree of permeability.

Lines 170-171: It is overstated. Please see (1) above.

Response 27: We have reworded this sentence (and other similar ones) to couch our description of OMVs as they pertain to our study and also flushed out in more detail why they are useful for the described approaches.

Figure 1B: Did the authors conduct any cryo-EM imaging to avoid the interference of sample preparation?

Response 28: We agree with the reviewer about the potential for artifacts in these types of approaches and have provided additional information to mitigate some of this concern.

First, in order to account for effects of fixation and sample preparation for EM, we compared the polymyxin B treated sampled to untreated (buffer-control) vesicles and well as PMBN treated samples, and did not observe vesiculation effects, indicating the effect is independent of the fixation and processing and was only apparent in the presence of polymyxin B (**SI Fig. S1**).

Second, as suggested by the reviewer, we performed cryo-EM on with wild-type and lipid A modified polymyxin-resistant OMVs (**SI Fig. S2**). Polymyxin B treatment of wild-type OMVs led to clumping (which we also previously observed by TEM). However, less or no OMV clumping was observed with PMBN or buffer treatment of wild-type OMVs. Moreover, outer membrane abnormalities were observed only in polymyxin B treated OMVs, though the contrast in the negative stained OMVs allowed for clearer visualization of these deformities (however, this image processing was not done on bulk images used for quantification). Thus, polymyxin B treatment specifically led to clumping and outer membrane distortion as observed by two methods. We

repeated these experiments on whole *E. coli* cells and observed membrane protrusions (as have been previously reported in the literature) only in the PMB-treatment of wt- *E. coli*.

As the physical roles for either of the noted cryo-EM observations in the actual permeabilization and killing activities of polymyxins are not currently known, we have withheld additional speculation but did, as requested by Reviewer 2, quantify the clumping to provide a quantitative metric for comparison. This is now described in the methods and supplemental sections and data are provided in **SI Fig. S2**. However, the inability to clearly image (by EM or Cryo-EM) the large clumps primarily found in the PMB treated wt-OMV samples means that quantification must be interpreted with the critical caveat that the images analyzed were not a random sample as large clumps had to be avoided.

Line 889: Should (C) be (D)?

Response 29: The reviewer is corrected. We have re-worked Figure 1 as suggested by another reviewer and remedied this mistake.

SI Table S1: The structures of different types of lipid A should be provided.

Response 30: We agree that this could help the reader. We have now included structures of lipid A and examples of polymyxin-resistant modified lipid A (**SI Fig. S5**).

SI Figure S4: The structure of PMBN is wrong and Dab1 should be removed.

Response 31: Many thanks to the reviewer for catching this. We inadvertently used the structure of the polymyxin B decapeptide. This has been corrected such that polymyxin B nonapeptide is now shown (**SI Fig. S7**).

Reviewer #2 (Remarks to the Author):

This is a significant and well done piece of work. It has shed great light on the mechanisms by which polymyxins interact with the outer membrane of bacteria to eventually cause bacteria cell death. The work is technically well done and uses appropriate controls. The authors have done a fine job in communicating their results clearly and objectively in the context of what is known in the current literature. I found it easy to read despite what could be more challenging biophysical and mathematical components to some readers. I think the study opens up new horizons to researchers in targeting bacterial cell membranes for enhancing molecules or new chemical entities to destroy bacterial cell membranes.

Response 32: We thank the reviewer for the helpful feedback and supportive comments.

I only have some minor points for clarification/improvement.

The EM data in Figure S1 is decent but the significance of changes induced by polymyxin are a little murky. I suggest a semiquantitative approach to presenting data from images. Perhaps, changes in OMVs per area or tubules per OMVs, etc. can be considered. Some changes are discerned from the images by eye but how prominent they are is harder to interpret.

Response 33: Similar points were raised by the other reviewers and we have copy-pasted our response here again for completeness.

We agree with the reviewer about the potential for artifacts in these types of approaches and have provided additional information to mitigate some of this concern.

First, in order to account for effects of fixation and sample preparation for EM, we compared the polymyxin B treated samples to untreated (buffer-control) vesicles and well as PMBN treated samples, and did not observe vesiculation effects, indicating the effect is independent of the fixation and processing and was only apparent in the presence of polymyxin B (**SI Fig. S1**).

Second, as suggested by the reviewer, we performed cryo-EM on wild-type and lipid A modified polymyxin-resistant OMVs (**SI Fig. S2**). Polymyxin B treatment of wild-type OMVs led to clumping (which we also previously observed by TEM). However, less or no OMV clumping was observed with PMBN or buffer treatment of wild-type OMVs. Moreover, outer membrane abnormalities were observed only in polymyxin B treated wild-type OMVs, though the contrast in the negative stained OMVs allowed for clearer visualization of these deformities. Thus, polymyxin B treatment specifically led to clumping and outer membrane distortion as observed by two methods. We repeated these experiments on whole *E. coli* cells and observed similar results. As requested by the reviewer, we performed quantification and present the data, but with significant limitations and caveats as discussed above (Response 28).

As the physical roles for either of the noted cryo-EM observations in the actual permeabilization and killing activities of polymyxins are not currently known, we have withheld additional speculation but did, for a matter of completeness, quantify the clumping to provide a quantitative metric for comparison. This is now described in the methods and supplemental sections and data are provided in **SI Fig. S2**.

How did the authors minimize refractive index change from buffer compatibility with increasing concentrations injected. Was any background refractive change apparent?

Response 34: The reviewer raises some useful points regarding these data. To minimize buffer effects, polymyxin B powder was brought up in the same buffer stock used for that run and for all subsequent dilutions of polymyxin B. We also referenced to a channel without loaded OMVs (or whole *E. coli* cells) but with the same concentration of polymyxin injected, and then referenced a second time to a preceding run with OMVs (or *E. coli* cells) loaded but with buffer alone injected. This double referencing was performed for all experiments. A solvent correction cycle (6 dilutions) was run before and after each run (if DMSO was present).

Reviewer #3 (Remarks to the Author):

The authors describe a study investigating how a class of antibiotic (polymyxins) interacts with a molecule called lipid A, which is found in the protective outer layer of several gram-negative bacteria. Cationic polymyxins are known to bind to negatively charged lipid A, resulting in a conformational change that ultimately leads to pores forming in the membrane and the killing of the cell. However much is still unknown about this process, and in order to understand it better, the researchers used surface plasmon resonance (SPR) to study the stages of interaction between polymyxins and lipid A. Second, they develop a mechanistic model to explain how transient binding of polymyxins to LPS facilitates incorporation into the bacteria, promoting phase separation and the emergence of high density clusters. They propose a mechanism by which killing of cells would be consistent with their SPR data, model, and previously reported investigations. This mechanism could be a useful insight for the development of other antibiotics targeting similar molecules, and could be an insight into how cells develop resistance to polymyxins by failing to get killed. The premise of the article is potentially an important contribution to the high priority field of antibiotics and antibiotic resistance. However, I have some concerns which I feel should be addressed before proceeding.

Response 35: We thank the reviewer for the thoughtful consideration of our manuscript and excellent summary.

Major

The authors propose a 6 parameter model that is then fit to the SPR data. The high number of parameters runs the risk of over-fitting. Usually simpler models (for example a 2 state model with 4 parameters in this case) should be tested first and found to be insufficient to explain the data before moving to more parameters. The authors should provide adequate quantitative justification for their model complexity, e.g. by showing why a simpler model would not suffice.

Response 36: The model from the first submission was intended as an approximate semi-empirical model and we fitted many simpler models before arriving at this three-state model. Examples of such approximate fits are shown in the new **SI Sections 1B-1G**. However, with addition of many orthogonal experiments and state-of-the-art modeling software (Kintek Explorer), we have been able to greatly improve our ability to iteratively optimize the model and have arrived at a model that follows mass action chemical kinetics. The rate of formation and interconversion of each physical state and the relative fraction of polymyxin B in each state are defined. However, the volume averaged SPR response does not permit discrete spatial resolution thereby preventing the number and size of clusters be estimated. We are confident that this functional model does reveal the fundamental processes that drive polymyxin B loading but the physical description of the system provided in **Fig. 4** is likely overly simplistic and will likely require refinement as a full physical understanding becomes available.

We have now added **Fig. 3** to the manuscript which provides a detailed account of how the three-state model was derived from multiple different experiments where each experiment was tailored to address specific questions. We provide full details of these experiments in our expanded supplemental section (see **SI Section 1B-1G**) and a full account of our interpretation. This also includes finite element analysis based modeling (**SI Fig. S13**) to validate our treatment of mass transport effects. Optimization did result in addition of a transition state (which remains at insignificant relative fractions(<0.1%)) that facilitates polymyxin B clustering via simple self-self interactions. We have added no additional term beyond what is specified by the reaction. When we fit our re-optimized three-state model to the data for polymyxin B binding to resistant-OMV data (**Fig. 5B**), it spontaneously returned near zero rate constants for state 3, effectively eliminating this state from the fit, while returning reasonable values for the other fitted parameters.

Thus, the quality of the data and modeling readily identified simplification and constraints that could be made. **Fig. 5** and **Table 2** clearly show that binding of polymyxin B to resistant-OMVs is dominated by transient interactions with LPS and slow formation of nucleates but unlike wt-OMVs these nucleates do not lead to formation of clusters. This mechanism is consistent with the ability of lipid A modifications that lead to polymyxin resistance to stabilize the LPS affinity network (e.g., Khondker et al. (2019) *Communications Biology*), even in the absence of metal ions, preventing membrane stretching which appears essential to polymyxin B clustering. We do intend to more broadly apply the model to characterize other compounds and experimentally connect our model to cell killing but this is beyond the scope of the current work.

The authors do not provide any form of systematic cross validation study for their model, i.e. to test its predictive capability in contexts other than the data used to parameterize the model. For example they could have used leave-one-out cross-validation or k-fold cross-validation, i.e. methods where the model is parameterized with a subset of the available data and then tested for validity against the remaining data.

Response 37: The main goal of our current modeling was to develop a mechanistic model that would allow a direct comparison of the rate constants for a quantitative comparison of how polymyxin B binding to resistant-OMVs differs from binding to wt-OMVs and this has now been achieved rigorously here. The model also provides a binding mechanism that is consistent with self-promoted uptake which will be explored but is beyond the scope of the current work

The parameters determined from fitting the model to SPR data shown in Table 2 are determined with the barest minimum of statistical rigor, with $n=2$, and there is no discussion of the type of replicates used – are these just technical replicates done on the same sample? Or, if I am not mistaken – it seems like 2 *different* concentrations were used, then averaged, and then this is called $n=2$?? If this is the case, then that does not make any sense! In table 2 for the second-to-last item KD cPMB, it looks as though the points 0.186 and 0.202 are both way outside the reported mean \pm standard deviation (0.194 \pm 0.01). Or are the replicates actually distinct experimental samples that were prepared independently? In any case, there appears to be insufficient transparency about the quality of statistical sampling, reproducibility, or shape of the distributions being investigated.

Response 38: We have now implemented full global analysis over three multi-dose titration curves, containing 13 injection segments, providing SE, CL for each parameter with a rigorous analysis of parameter bounding and uniqueness (2D Fitspace). The 2D fitspace associated with **Fig. 5A** and **5B** plots (right panels) show (using color contrast) parameter bounding and covariance for all paired combinations of parameters. All parameters plots show a distinct boundary in all directions indicating well constrained parameters. A 2D FitSpace boundary with greater elliptical form indicates a degree of parameter correlation with wider confidence intervals while broken boundaries that continue unbounded in any given direction indicate unconstrained parameters, which is unacceptable. The fitted parameter values are shown in **Table 2** along with the associated confidence intervals.

The large number of model parameters also creates the risk for degenerate solutions – where multiple sets of very different values of the rate constants could achieve equally good fits. The authors should provide evidence that their model is sufficiently constrained such that this is not the case – either by fitting to a subset of the data for example and showing that parameter values do not change drastically from their current form. Or by conducting more experimental replicates and showing that the distribution of parameter values is well-behaved - i.e. normally distributed around a single value.

Response 39: Please also see our Response 35 above. We did fit simpler models that proved mechanistically unreliable since they would generally fit data over very specific concentration regimes. This investigative experimental work was not included in the initial submission of this manuscript as it was considered too lateral to our findings but we have now expanded the supplemental to include such detail. A key element of our modeling was the use of orthogonal SPR experiments to inform on constraints that could be adopted. For example, a boundary layer model (*Eqn (S3)*) provided good estimates for reversible binding of PMBN to LPS (**Fig. 3A** and **3B**). Briefly, *Eqn (S3)* was fit to this data using Biaevaluation (leading SPR-analysis software) where the initial guess for K_D was obtained by first fitting an affinity model to the steady-state regions of these same data set. k_t was calculated from theory (Goldstein et al. (1999) *Journal of Molecular Recognition*) and a global fit returned values that were then used as starting values for the three-state model fit of PMBN binding to LPS (**Fig. 5**). Kinetic constants for state 3 tended to zero while the other fitted parameters remained resolved. Thus state 3 was eliminated from the model by the fitting algorithm resulting in the fit in **Fig. 5B**. The three-state model was then fit to the polymyxin B data where initial parameters for k_t , K_{D1} , k_3 , and k_4 were taken from the PMBN fit. The simulation did not resemble the data set and therefore the initial values for the added kinetic rate constants associated with the transition state and state 3 were iterated manually. This was performed using Kintek's dynamic simulation function where one can drag the value of any parameter over a wide range of values in just a few seconds and the simulation, which is overlaid on the actual curves to be fit, responds in real time (no apparent delay) such that the observed curvature and scaling of the binding curves to be fit is roughly reproduced in the simulated curves and requires just a few minutes of manual iteration. With these initial values a global fit showed some instability due to the higher number of parameters being estimated. Therefore, K_{D1} was held constant at the value taken from the affinity fit in **SI Fig. S14**. The fit was repeated and it was noted that k_3 and k_4 were repeating as k_6 and k_{10} , respectively, and this removed another two rate constants leaving just three binding constants and k_t to be estimated. A fit with these constraints produced a high quality fit as shown in (**Fig. 5A**) where initial values were found by manual dynamic simulation and repeating this fitting process multiple times had no significant effect on the results.

One of the major claims in this paper is that the modified version of polymyxin B (PMBN) lacks the long term membrane accumulation effect of polymyxin B. While the authors show the accumulation effect of polymyxin B on whole *E. coli* cells, and unless I am mistaken, they do not show how *E. coli* cells respond in the presence of PMBN. This seems like an important experiment to include to support the primary claim of this paper and the validity of OMVs as a model system in this context, especially considering how artificial the model system is, using tween-80 for example which is likely to affect lipid properties.

Response 40: The reviewer is correct that this is an important experiment that should be included to both demonstrate how PMBN interacts with cells and to support the OMVs as a model system for understanding polymyxin binding. These data are now included (**SI Fig. S8** and **SI Fig. S12**) and discussed in the revised manuscript. Our observations with cells were nearly identical to our observations with OMVs for PMBN, again reinforcing that the OMVs are useful surrogates for studying outer membrane interactions using SPR. We also investigated whether addition of tween-80 could be impacting binding. Briefly, experiments investigating binding of polymyxin B and PMBN to OMVs were conducted with, and without, the addition of tween-80 and we observed that the binding curves were near superimposable at high nM concentrations, indicating the absence of any interference. However, there was a systematic loss in the concentration of the polymyxins at the lower nM concentrations in the absence of tween-80 indicating that the only

observable effect of addition of tween-80 was to block non-specific retention sites along the injection tubing thereby preventing depletion of polymyxins before arrival at the SPR flow cell.

Fortunately, we also performed experiments with a non-specific LP chip that was incompatible with tween-80. These experiments show that we can measure similar binding profiles in the absence of tween-80 (**SI Fig. S4F**), leading us to conclude that the conditions we used did not perturb the binding we observed. This LP system is less ideal due to less robust regeneration and high background binding that reduced the ability to see binding at low doses which is why we focused our efforts on the C1 chip.

The authors claim that TEM imaging of outer membrane vesicles shows how polymyxin causes vesiculation and tubulation. However they use fixation/drying (as opposed to cryo-EM for example), so isn't this an approach that is likely to introduce artifacts? Likewise for the SEM image in Fig 1C – it is hard to say whether this is representative of intact surface coupling of aqueous-liquid-packed spherical vesicles given that these samples must be deposited, dried, and sputter coated with a layer of metal before imaging. Are there other articles to support these dry methods as a valid technique for characterizing lipid vesicles? References 25 and 62 cited for these steps respectively do not seem to show this.

Response 41: Similar points were raised by the other reviewers, and we have copied our response here again for completeness.

We agree with the reviewer about the potential for artifacts in these types of approaches and have provided additional information to mitigate some of this concern.

First, in order to account for effects of fixation and sample preparation for EM, we compared the polymyxin B treated sampled to untreated (buffer-control) vesicles and well as PMBN treated samples, and did not observe vesiculation effects, indicating the effect is independent of the fixation and processing and was only apparent in the presence of polymyxin B (**SI Fig. S1**).

Second, as suggested by the reviewer, we performed cryo-EM on with wild-type and lipid A modified polymyxin-resistant OMVs (**SI Fig. S2**). Polymyxin B treatment of wild-type OMVs led to clumping (which we also previously observed by TEM). However, less or no OMV clumping was observed with PMBN or buffer treatment of wild-type OMVs. Moreover, outer membrane abnormalities were observed only in polymyxin B treated wild-type OMVs, though the contrast in the negative stained OMVs allowed for clearer visualization of these deformities. Thus, polymyxin B treatment specifically led to clumping and outer membrane distortion as observed by two methods. We repeated these experiments on whole *E. coli* cells and observed similar results.

As the physical roles for either of the noted cryo-EM observations in the actual permeabilization and killing activities of polymyxins are not currently known, we have withheld additional speculation but did, for a matter of completeness, quantify the clumping to provide a quantitative metric for comparison. This is now described in the methods and supplemental sections and data are provided in **SI Fig. S2**.

In methods, the authors state: "Sensorgrams could not be normalized for bound ligand (i.e., the OMVs on the chip surface) using a calculated 'Rmax', and, as such, analyzed traces, including those in the figures, were selected to have similar levels of loaded OMVs." Does this mean that the authors manually selected which runs to include? How many runs were discarded in this process? What did they look like? Such a practice is going to be prone to human selection bias –

the proper way to do with would be to choose an objective formal selection criterion and state it along with the actual variation in bound ligand between samples that were included.

Response 42: We can understand the concerns from the reviewer here as we did not provide sufficient information to fully understand our intent, but now address it fully. The need to record curves with matching coating levels is entirely related to maintaining global fitting constraints, which are essential to robust model fitting. Ordinarily, we expect replicated sensorgrams recorded at different R_{max} levels to be superimposable after response-normalization. However, there can be differences between these normalized binding curves at higher R_{max} levels because higher R_{max} levels increase the degree of mass transport limitation which tends to slow the observed binding kinetics. However, we were aware that mass transport limitation was entirely dominant for polymyxins binding to LPS because we could see that the observed kinetic curvature for binding of polymyxins to LPS was extraordinarily sensitive to R_{max} . Therefore, in order to maintain our ability to obtain a global fit value for R_{max} and k_t over a set of binding curves it was critical to run replicates that were recorded using the same target-coating level. Failure to maintain this would have resulted in local fitting of both R_{max} and k_t which would introduce an additional four fitted parameters, destabilizing the model significantly. We have performed extensive theoretical and experimental studies (now added to supplemental **SI Section 1**) and have confirmed that mass transport limitation is entirely dominant for polymyxin B binding to LPS-rich coatings. Such binding is rarely observed, as it requires a very high reaction flux $L_r = k_t \cdot R_{max}$, but this is unavoidable here due to the combination of rapid, electrostatically dominated binding of polymyxin B to densely packed LPS surfaces. In this regime, a concentration gradient in polymyxin develops above the sensing surface upon the start of an injection, and decays at the end, resulting in transport-driven kinetic curvature in the sensorgrams. In this regime, the observed kinetics vary as a function of LPS density (i.e., varies with R_{max}) and affinity (K_{D1}) rather than the binding kinetics of the polymyxins to LPS. However, the ordinary differential equation shown in *Eqn (S3)*, allows the pseudo-steady-state occupancy at any time, the binding affinity constant and the R_{max} to be determined under these unusual conditions (see **Fig. 3A, 3B**, and **SI Fig. S12A-S12C**).

Minor

The lettering order in Fig 2 is confusing, with order going columnwise and then rowwise in the same panel.

Response 43: We have remade **Fig. 2** to remedy the confusing lettering order and included additional data.

Fig 1 D and E are a plot of the same data, and it is unnecessary to show it twice. The annotation could be added to the first plot to communicate the same effect, and it would be less misleading as it currently looks at first glance like two different experiments.

Response 44: We thank the reviewer for this point. Our original thought was to provide these data twice to illustrate different points, but agree that it is unnecessarily redundant and have reduced this to a single figure (**Fig. 1D**) as suggested.

Polymyxins are known for exhibiting nephrotoxicity, and the mechanism for this toxicity is thought to involve binding to phospholipids in the cell membrane. Could the authors comment on any similarities or differences between human versus bacterial cell toxicity and any insight their model might or might not provide here – for example regarding differences or similarities in membrane organization?

Response 45: The reviewer is correct that nephrotoxicity is a known liability of polymyxins. Although we have not addressed this point directly in the current work, we have included data capturing the interaction of polymyxin B with mammalian cell derived vesicles (**SI Fig. S4A** and **S4C**), which is distinct from what we observe with OMVs and bacterial cells (**Fig. 2A** and **2C**). This provides a potential starting point for future investigations into polymyxin toxicity, including asking how polymyxins interact with renal proximal tubule epithelial cells, how they accumulate in these cells, and what the mechanism of non-lipid A mediated toxicity is.

Reviewer #3 (Remarks to the Author):

The authors have addressed my points adequately, having done additional experiments and having added more rigorous analysis and exploration of their model. I believe the manuscript is suitable for publication.

Reviewer #4 (Remarks to the Author):

After carefully reviewing both the manuscript and the point-to-point response, I believe the authors have properly addressed the reviewers' comments. This includes: 1) Measured the interaction of PMB and PMBN with *E. coli* bacterial cells and the purified LPS in addition to OMVs; 2) Performed supportive experiments in Fig. 3 for the three-state model; 3) Introduced Kintek modelling to optimize their model and obtained consistent results; and 4) Implemented full global analysis for better data fitting. Apparently, the revised version is much improved compared to the first submission. I have no further comments with the SPR technique and the mathematical modelling which are out of my research expertise. However, with the limitation of the approach, I am afraid it might be difficult to make precise conclusion in order to answer how polymyxins kill bacterial cells. Here are my specific comments surrounding the biological part:

1. In Fig. 2 and the related text, the authors measured SPR response between PMB/PMBN and resistant OMVs but did not mention the resistance mechanism. Could the authors clarify if the resistance was due to *pmrA* or *mcr-1* since both mutants of *E. coli* were studied?
2. It is known that mutations in *pmrA* and *mcr-1* may cause different lipid A modifications (e.g., 4-amino-4-deoxy-L-arabinose and phosphoethanolamine) at different proportions (SI Table S1) and lead to different resistance levels against polymyxin B as observed in Table 1. It would be interesting to compare the interaction difference between the *pmrA*-OMVs and the *mcr-1*-OMVs with PMB.
3. Based on *m/z* calculation, SI Fig. 6 only contains phosphoethanolamine modified lipid A. This is not consistent with SI Fig. 5. The authors need to clarify how these lipid A structures been obtained.
4. The calculation of binding stoichiometry in Response 24 is not well justified. First, the rough-LPS used in this study is a Rd mutant from *E. coli* F583, which is much shorter than Ra type. According to the structure, the average molecular weight of Rd-LPS should be ~2000 Da rather than 3500 Da which is more reasonable for the longer Ra-LPS. Second, if the authors would like to build a link between the molecular weight of LPS and the SPR response, they should use different types of LPS, such as smooth-LPS, different rough-LPS, and lipid A to perform the binding assay. Moreover, modified-lipid A extracted from the *pmrA* and *mcr-1* mutants of *E. coli* should be considered using in the study as well.

We thank the reviewers for their time and constructive comments. We have provided a point-by-point response to the issues raised and updated our text to address the reviewer concerns. We have also added the following to the manuscript:

- A new figure (SI Fig. S4I) to allow for a direct comparison between cells with different polymyxin-resistant lipid A modifications
- Added detail to describe the high-resolution mass data we used to assign modifications to lipid A species (SI Fig. 6 figure legend) as well as a more detailed reporting of the modifications observed (new SI Table S2)
- Re-analysis of the binding stoichiometry using the appropriate MW as well as refractive indices for the lipid A species used in our experiments

We hope that the reviewers agree that these changes alleviate any remaining concerns about this foundational study and look forward to reporting these findings to the scientific community.

REVIEWER COMMENTS

Reviewer #3 (Remarks to the Author):

The authors have addressed my points adequately, having done additional experiments and having added more rigorous analysis and exploration of their model. I believe the manuscript is suitable for publication.

We appreciate the time and effort of the reviewer in reading and reviewing our revised manuscripts.

Reviewer #4 (Remarks to the Author):

After carefully reviewing both the manuscript and the point-to-point response, I believe the authors have properly addressed the reviewers' comments. This includes: 1) Measured the interaction of PMB and PMBN with E. coli bacterial cells and the purified LPS in addition to OMVs; 2) Performed supportive experiments in Fig. 3 for the three-state model; 3) Introduced Kintek modeling to optimize their model and obtained consistent results; and 4) Implemented full global analysis for better data fitting. Apparently, the revised version is much improved compared to the first submission. I have no further comments with the SPR technique and the mathematical modelling which are out of my research expertise. However, with the limitation of the approach, I am afraid it might be difficult to make precise conclusion in order to answer how polymyxins kill bacterial cells.

We thank the reviewer for their time in reading our manuscript and for the constructive feedback. Our model establishes that the accumulation of polymyxin clusters is a necessary step in cell killing but does not specify steps that define the cell killing mechanism as there could be additional steps not revealed in our approach. The value of our model is in providing a basis for understanding the mechanistic differences that drive the biological activity of polymyxin B relative to the non-cell killing polymyxin B

nonapeptide and relative to polymyxin-resistant strains, something that had not previously been achieved. Precisely how polymyxin B clustering in the outer membrane induces cell killing remains to be elucidated. To avoid confusion, we have adjusted our abstract and made sure to indicate where we are speculating about the killing mechanism in the discussion and how this is one critical step in the process. We hope this addresses the concern.

Here are my specific comments surrounding the biological part:

1. In Fig. 2 and the related text, the authors measured SPR response between PMB/PMBN and resistant OMVs but did not mention the resistance mechanism. Could the authors clarify if the resistance was due to *pmrA* or *mcr-1* since both mutants of *E. coli* were studied?

Thank you for pointing out this omission. We have noted in the Figure 2 legend and in the text the source of the resistant strains used for these experiments. Also, see below for a comparison of these two modification mechanisms.

2. It is known that mutations in *pmrA* and *mcr-1* may cause different lipid A modifications (e.g., 4-amino-4-deoxy-L-arabinose and phosphoethanolamine) at different proportions (SI Table S1) and lead to different resistance levels against polymyxin B as observed in Table 1. It would be interesting to compare the interaction difference between the *pmrA*-OMVs and the *mcr-1*-OMVs with PMB.

The reviewer is indeed correct that the types and proportions of lipid A modifications were variable. In our experiments, the phosphoethanolamine modification predominated (see our new SI Table S2 which adds a breakdown of the lipid A species by specific modification and resolves the previous disconnect). In the interactions we describe, we did not find any features of binding that could be ascribed to, or explained by, the differing percentages of modification in the OMV-batches. We also note that SPR on whole cells, which we demonstrated have the same kinetics as OMV and purified LPS binding, showed similar responses for strains with *PmrA*^{G53E} and carrying *p-mcr-1*. That being said, we do not rule out that variable types or levels of modification could affect polymyxin binding.

We also note that our approach has not revealed any significant differences between polymyxin B binding to *E. coli* *PmrA*^{G53E} and *E. coli* *pmcr-1*. We have added an additional figure (SI Fig. S4I) measuring binding of polymyxin B to *E. coli* *PmrA*^{G53E} cells to allow direct comparisons with our reported binding to the *E. coli* *p-mcr-1* cells (Fig. 2D). It will be interesting to explore binding to more 'natural' modified strains that are polymyxin resistant but not locked into their phenotype by engineered mutations to determine the spectrum of binding by polymyxins that might be expected in a clinical setting, but this was left for future directions.

3. Based on m/z calculation, SI Fig. 6 only contains phosphoethanolamine modified lipid A. This is not consistent with SI Fig. 5. The authors need to clarify how these lipid A structures been obtained.

We thank the reviewer for pointing this out. High-resolution, accurate mass data was acquired and analyzed for pEtN and Ara4N-modified lipid A analogues. The associated multiply charged ion was extracted from the total ion chromatogram and integrated to give the relative abundance of modified and unmodified lipid A. The structures were based on literature structures and accurate mass data (Aghapour et al. 2019. Infect Drug Resist and Knopp et al. 2021. Plos Genet). SI Fig. 6 is pEtN integrated XIC chromatograms and are representative of how L-Ara4N and pEtN modification LCMS data were processed. Importantly, in the revision, we have clarified this by extensively revising the SI Fig. 6 figure legend and hope this remedies the confusions.

4. The calculation of binding stoichiometry in Response 24 is not well justified. First, the rough-LPS used in this study is a Rd mutant from E. coli F583, which is much shorter than Ra type. According to the structure, the average molecular weight of Rd-LPS should be ~2000 Da rather than 3500 Da which is more reasonable for the longer Ra-LPS. Second, if the authors would like to build a link between the molecular weight of LPS and the SPR response, they should use different types of LPS, such as smooth-LPS, different rough-LPS, and lipid A to perform the binding assay.

We thank the reviewer for pointing out the mistaken LPS MW used in our calculations. We have confirmed that the MW of the species used in our assays has an average range of 1.7-1.8 kDa. This new MW has been used to redo the calculations (see revised SI Section 1G. Note that this resulted only in a minor change in the results (PMB:LPS ratio 0.45:1 which is equivalent to an LPS:PMB ratio 2.2:1) and this does not change our overall conclusions that clustering can lead to super-stoichiometric accumulation.

Moreover, modified-lipid A extracted from the pmrA and mcr-1 mutants of E. coli should be considered using in the study as well.

We agree that this could be a future direction. In the current work we aimed to provide a foundational model for understanding the interactions of polymyxins with the outer membrane and we hope this inspires future experiments to elucidate how our mechanistic model couples with other processes that drive the complete mechanism of activity and resistance for polymyxins and other lipid A-binding molecules. Practically, however, our results with OMVs and whole cells demonstrate that these platforms capture all of the relevant kinetic measurements and thus serve as useful surrogates for determining the polymyxin binding mechanism, which is especially useful in situations where high-quality, commercially available reagents, such as modified LPS, are not available.

Reviewer #4 (Remarks to the Author):

After carefully reviewing the revised manuscript, I believe the authors have addressed my comments properly. I have no further comments.